# Prescriptive Scaling Reveals the Evolution of Language Model Capabilities

**Hanlin Zhang** [* 1]   **Jikai Jin** [* 2]   **Vasilis Syrgkanis** [2]   **Sham M. Kakade** [1]

## Abstract

Machine learning model performance improvements tend to arise from competition and application. For deployment, we consider *prescriptive scaling laws*: given a pre-training compute budget, what downstream accuracy is *attainable* with contemporary post-training practice, and how stable is that mapping as the field evolves? Using large-scale observational evaluations with 5k existing and 2k newly evaluated model checkpoints spanning 2022–2026 across six benchmarks, we estimate *capability boundaries*—high conditional quantiles of benchmark scores as a function of log pre-training FLOPs, via smoothed quantile regression with a monotone, saturating sigmoid parameterization. We validate temporal reliability by fitting on earlier model generations and evaluating on later releases: across four of six tasks the out-of-distribution coverage error remains below 2%, while math reasoning exhibits a consistently advancing boundary over time. For instance, at a budget of $10^{24}$ FLOPs the estimated attainable accuracies are 0.83 on IFEval and 0.54 on MATH Lvl 5. We then extend our approach to analyze task-dependent saturation and to probe contamination-related shifts on math reasoning tasks. Finally, we introduce a balanced I-optimal sampling algorithm that recovers near-full-data frontiers using roughly 20% of the parameter-count-weighted evaluation budget (as low as 5% on some tasks) while maintaining comparable calibration. Together, our work releases the Proteus-2k, the latest model performance evaluation dataset, and introduces a practical methodology for translating compute budgets into reliable performance expectations and for monitoring when capability boundaries shift across time.

---

[*]Equal contribution [1]Harvard University, Cambridge, MA, USA [2]Stanford University, Stanford, CA, USA. Correspondence to: Hanlin Zhang <hanlinzhang@g.harvard.edu>, Jikai Jin <jkjin@stanford.edu>.

*Proceedings of the 43rd International Conference on Machine Learning*, Seoul, South Korea. PMLR 306, 2026. Copyright 2026 by the author(s).

## 1. Introduction

Over the past several years, language model (LM) scaling has emerged as one of the most robust empirical laws in modern machine learning (Hestness et al., 2017). Across model families and training regimes, increasing pre-training compute has been shown to produce smooth and predictable improvements in loss, perplexity, and, to a lesser extent, downstream task performance (Brown et al., 2020; Hoffmann et al., 2022; Chowdhery et al., 2023; Gadre et al., 2024). This observation has driven a paradigm in which scale itself becomes a primary design variable, enabling practitioners to trade off data, model size, and compute in a principled way (Kaplan et al., 2020; Hoffmann et al., 2022).

As language models transition from research artifacts to deployed systems, the limitations of existing scaling laws have become increasingly pronounced. Despite the success of scaling laws, they do not answer one question that practitioners routinely face: *given a fixed pre-training compute budget $C$, what downstream performance can one realistically expect to achieve with high probability after post-training?* While average trends with respect to compute are sometimes stable, downstream behaviors of interest (such as reasoning performance, instruction following, or domain-specific question answering) exhibit substantial heterogeneity even among models trained with similar FLOPs (Jin et al., 2025). Post-training procedures (Ziegler et al., 2019), data curation choices (Setlur et al., 2024), and temporal effects (Dominguez-Olmedo et al., 2024) further complicate the relationship between pre-training compute and deployed performance, weakening the direct applicability of standard scaling laws for real-world decision making.

Recent work has highlighted this gap from multiple perspectives: downstream benchmark scaling can be noisy, benchmark-dependent, and weakly coupled to pre-training loss, in part due to heterogeneous training factors (e.g., data mixture, architectures, and evaluation artifacts) and the disconnection between loss and downstream accuracy (Lourie et al., 2025; Gadre et al., 2024; Chen et al., 2024; Schaeffer et al., 2024; Zhang et al., 2025a; Qi et al., 2025). At the same time, the rapid growth of public evaluation repositories—especially leaderboards that aggregate thousands of *post-trained* checkpoints—makes it increasingly feasible to study these relationships empirically from observational data.

| Benchmark | IFEval | BBH | MATH Lvl 5 | GPQA | MUSR | MMLU-PRO |
|---|---|---|---|---|---|---|
| **Acc. @ $10^{24}$ FLOPs** | 0.828 | 0.700 | 0.539 | 0.424 | 0.535 | 0.563 |

*Table 1.* Estimated attainable accuracies predicted by the no-split 0.98-quantile sigmoid boundaries at $10^{24}$ FLOPs.

In this paper, we study *prescriptive scaling*: given a base-model pre-training compute budget, what *attainable* post-training performance should we expect on a target benchmark? Rather than modeling only mean trends, we summarize the attainable region with *capability boundaries*: for each task we estimate a high conditional quantile of observed post-trained accuracy as a function of log pre-training compute (Koenker & Bassett, 1978). This framing is robust to outliers and recipe-specific variation, and it yields an end-to-end, decision-oriented compute-to-performance map from large collections of heterogeneous checkpoints. Crucially, we treat *time* as a first-class axis: by fitting boundaries on earlier model generations and validating on later ones, we can gain knowledge of whether a compute-based boundary remains predictive as training recipes and post-training techniques evolve.

We rely on three complementary data sources: (i) the Open LLM Leaderboard v1 (Beeching et al., 2023) and v2 (Fourrier et al., 2024), each containing thousands of models evaluated on six benchmarks under consistent metrics, (ii) public leaderboards for state-of-the-art frontier models (e.g., Epoch AI and LifeArchitect.AI), and (iii) newly added 2.4k open-weight models (PROTEUS-2K) focusing on releases from April 2024 through March 2026, including those after the Open LLM Leaderboard v2 retirement (2025-03-13), that we evaluate ourselves (including new model families of Qwen3 (Yang et al., 2025), Gemma-3 (Team et al., 2025), GPT-OSS (Agarwal et al., 2025)), following the same Open LLM Leaderboard pipeline. Together, these sources provide both breadth (many heterogeneous post-training pipelines) and a basis for assessing temporal validity (Dominguez-Olmedo et al., 2024). The main contributions are summarized below:

- **Sigmoid capability boundaries:** Compared with pre-trained model performance, we show that the attainable *post-trained* performance is much more predictable (OOD calibration error of 2.2% for the sigmoid boundary vs. 3.6% for a compute-agnostic baseline) and is well-characterized by a simple monotone, saturating sigmoid function of log-compute.

- **Temporal validity and task-dependent ceilings:** Using chronological train/validation splits, we find that capability boundaries for four of six tasks are comparatively stable over time, yielding a nearly deterministic relationship between compute and attainable accuracies, while math reasoning exhibits a consistently improving boundary. As an illustration, Table 1 provides estimated attainable accu-

racies (0.98-quantile sigmoid boundaries) at a budget of $10^{24}$ FLOPs.

- **Case studies: saturation and contamination:** We apply prescriptive scaling to revisit two evaluation issues. Our saturation analysis suggests two qualitatively different limits to "scaling": some tasks quickly hit a stable, size-determined ceiling, while others (notably math) exhibit an evolving ceiling over time. Our contamination analysis on frontier models finds no clear evidence of AIME-2025 score inflation due to contamination.

- **Efficient prescriptive scaling via adaptive sampling:** We propose a sampling algorithm that accurately recovers sigmoid capability boundaries under limited computation budget (typically $\approx 20\%$ of the full, parameter-count-weighted evaluation budget; and $\approx 5\%$ on some tasks).

## 2. Estimation of Post-training Capability Boundaries

Recent frontier-model reports (Achiam et al., 2023) emphasize an engineering goal of *predictable scaling*: using compute as a controllable input to forecast key training statistics and downstream benchmark behavior from smaller-scale runs, so that model development can be budgeted and planned in advance. Adopting this perspective, we use the term *Prescriptive Scaling* to denote the prescriptive question at the center of this paper: how a pre-training FLOPs budget translates into the range of targeted downstream performance attainable after standard post-training.

**Definition 2.1** (Prescriptive Scaling)**.** Given a certain budget of FLOPs $C$, the goal of *Prescriptive Scaling* is to train from scratch a model end-to-end to achieve a targeted downstream performance $y$ on a given benchmark.

Prescriptive Scaling, in the sense of budgeting training compute to reach a desired downstream behavior, is ultimately an engineering question: *given a resource constraint, what performance can one reliably attain with contemporary training and post-training practice?* For language models, the most consistently reported and directly controllable resource is pre-training compute. At the same time, deployed models are rarely raw checkpoints: they are produced by heterogeneous post-training pipelines (instruction tuning, RL, domain adaptation), and their benchmark scores exhibit substantial variance even at similar compute (Zhang et al., 2025b; Jin et al., 2025; Jiang et al., 2025).

To connect this broad "engineering for predictability" goal to measurable evidence, we narrow the problem to estimat-

ing *capability boundaries*: for each task, we ask how high the performance distribution of post-trained models reaches as a function of the base model's pre-training FLOPs. *This abstraction does not claim compute is the only driver.* Rather, we treat compute as a practical design coordinate. The estimated boundaries constitute empirical envelopes conditioned on the prevailing post-training methodologies, data curation practices, and evaluation protocols within the observed model ecosystem, thereby enabling the translation of a target accuracy level into a plausible range of computational requirements in a data-driven manner.

### 2.1. Setting and Modeling Assumptions

For each task, we collect evaluation results for a set of *post-trained* models. Each observation $i$ is a model paired with (i) an estimated *base-model* pre-training compute budget $C_i > 0$ (FLOPs) and (ii) an observed score $y_i \in [0, 1]$. Multiple models can share the same $C_i$ when they are derived from the same base model. We work in log-compute $z_i = \log_{10} C_i$. When assessing temporal generalization, we further group observations into chronological periods $\mathcal{P}_1, \ldots, \mathcal{P}_4$ (spanning mid-2022 through March 2025) and fit on one period at a time. Motivated by the practical need for setting a computational budget for a targeted accuracy, we treat base-model pre-training compute as the primary conditioning variable for attainable post-training capability. We are interested in the mapping

$z \mapsto$ attainable (upper-tail) accuracy of post-trained models with log-pretraining-compute $z$,

and we refer to this mapping (at a fixed quantile level $\tau$) as a *capability boundary*. One major challenge towards this goal is that outliers are ubiquitous across all model families.[1]

So, rather than studying the genuine maximal accuracy that we observe from evaluation results, we instead focus on recovering $q_\tau(z) \approx Q_\tau(Y \mid Z = z)$, the conditional $\tau$-quantile of the observed accuracy $Y$ given log-pretraining-compute $Z = z$. Note that $q_\tau(\cdot)$ should be read as an empirical attainable boundary for the observed model population. As with any observational study, if an underrepresented model family or recipe class consistently achieves higher scores at fixed compute, then the true attainable boundary could lie above our estimate; conversely, the main use case of prescriptive scaling is to provide a conservative, decision-oriented compute-to-performance map that can be updated as new families and recipes enter the evaluation ecosystem.

To estimate $q_\tau(z)$, we approximate it with a parameterized estimator $q_\tau(z; \theta)$ where $\theta$ is a learnable parameter. Define $\hat{y}_i = q_\tau(z_i; \theta)$ and minimize a smoothed pinball loss, a standard objective for quantile regression (Koenker, 2005;

---

[1]See Section B.1 for a concrete illustrative example and a brief discussion.

Narayan et al., 2024; Steinwart & Christmann, 2011):

$$\mathcal{L}(\theta) = \sum_{i \in \mathcal{P}_t} \ell_\tau(y_i - \hat{y}_i) + \lambda \, \Omega(\theta),$$
$$\ell_\tau(u) = \tfrac{1}{\kappa} \log(1 + e^{\kappa u}) + (\tau - 1)u.$$

We use $\tau = 0.98$, $\kappa = 50$, $\lambda = 10^{-3}$. Sensitivity analyses are performed in Section G.

### 2.2. Capability Boundary Estimators

For each task and training period, we fit a function $q_\tau(z)$ intended to approximate the conditional $\tau$-quantile $Q_\tau(Y \mid Z = z)$. We compare the following classes:

- **Constant (no-compute) baseline:** $q_\tau^{\text{const}}(z) = c$, a single scalar for all $z$.

- **Binwise constant:** partition $z$ into $B$ bins with edges $e_0 < \cdots < e_B$ computed *from the training z-values only*. Predict $q_\tau^{\text{bin}}(z) = c_b$ for $z \in [e_b, e_{b+1})$ (last bin inclusive), with $b = 0, \ldots, B - 1$.

- **Sigmoid:** a monotone saturating function in $z$, in the form of $q_\tau^{\text{sig}}(z; \theta) = y_0 + L\,\sigma(a + \beta z)$, $\sigma(t) = \frac{1}{1+e^{-t}}$, with $\beta \geq 0$, $0 \leq y_0 \leq 1$, and $0 \leq L \leq 1 - y_0$.

- **I-spline:** a strictly more general function class than sigmoid (a flexible monotone baseline), where we replace the linear predictor $a + \beta z$ with a monotone spline and pass it through a sigmoid so predictions remain saturating in $[0, 1]$. (Full definition and constraints are given in Section B.3.)

**Bin construction for the binwise model.** We use group-aware equal-mass binning on the training $z$-values only, never splitting identical $z$ values across bins. The full boundary-placement and minimum-bin-size merging procedure is provided in Section B.2; the resulting edges $e_0 < \cdots < e_B$ are used for both training and evaluation.

### 2.3. Evaluation Metrics

We report two complementary error metrics:

1. **Pinball loss (quantile accuracy).** We evaluate the mean smoothed pinball loss on both train and OOD validation periods. This is a proper scoring rule for quantiles in the unsmoothed limit and directly reflects how well $q_\tau(z)$ targets the $\tau$-quantile under asymmetric penalties (underprediction is penalized more heavily when $\tau$ is close to 1). The main limitation is that as a scalar aggregate, it can hide where errors occur (e.g., at low vs high compute) and its sign (underestimate vs. overestimate), which motivates us to include an extra coverage metric.

2. **Coverage error.** Within each log-compute bin $[e_b, e_{b+1})$, let $\mathcal{I}_b = \{i : z_i \in [e_b, e_{b+1})\}$ and $n_b = |\mathcal{I}_b|$. We compute empirical coverage $\hat{\tau}_b = \frac{1}{n_b} \sum_{i \in \mathcal{I}_b} \mathbf{1}\{y_i \leq \hat{y}_i\}$

and report the signed deviation $\hat{\tau}_b - \tau$. This measures whether the fitted capability boundary achieves the intended quantile coverage locally in compute.

# 3. Sigmoid Scaling Laws for Post-training Performance Boundaries

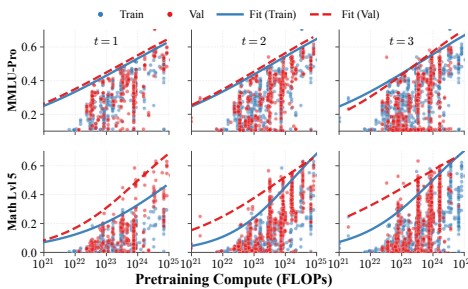

*Figure 1.* **Sigmoid capability boundaries across time.** In each subfigure, points correspond to post-trained models (x-axis: base-model pre-training compute; y-axis: benchmark score). We compare sigmoid fits across consecutive periods $(\mathcal{P}_t, \mathcal{P}_{t+1})$ for $t = 1, 2, 3$, visualizing both (i) the boundary fit on $\mathcal{P}_t$ and (ii) the boundary fit on $\mathcal{P}_{t+1}$ to illustrate boundary shift.

We now apply this methodology to characterize post-training capability boundaries, beginning with open-weight models on the Open LLM Leaderboard (Section 3.1), then probing external validity, and finally connecting to the classical pre-training scaling-law perspective (Section 3.2).

## 3.1. Cross-temporal Scaling of Open-weight Models

In this subsection, we study open-weight models on the Open LLM Leaderboard. To stress-test temporal generalization, we partition all models into four chronological evaluation periods: $\mathcal{P}_1$ ($\leq$2024-06), $\mathcal{P}_2$ (2024-07 to 2024-09), $\mathcal{P}_3$ (2024-10 to 2024-12), and $\mathcal{P}_4$ (2025-01 to 2025-03); see Section D for per-period counts. We then evaluate three rolling train–test pairs $(\mathcal{P}_t, \mathcal{P}_{t+1})$ for $t \in \{1, 2, 3\}$: for each $t$, we fit the $\tau$-capability boundary on $\mathcal{P}_t$ and evaluate out-of-distribution on $\mathcal{P}_{t+1}$, restricting evaluation to the overlap of the train and OOD ranges in $z$ to avoid extrapolation.

We focus on two questions: which function class best captures the observed compute–performance boundary, and how the fitted boundaries drift over time. Prior work has explored alternative function classes primarily for pre-training scaling laws (Caballero et al., 2023; Donoway et al., 2025); here we evaluate these alternatives in the post-training regime. On the other hand, while temporal effects can inflate pretrained models' benchmark scores (Dominguez-Olmedo et al., 2024), a large-scale study that incorporates post-trained models is still lacking.

### 3.1.1. THE SHAPE OF CAPABILITY BOUNDARY

In Section 2.2 we discussed several candidate estimators for the $\tau$-capability boundary. Table 2 reports the in-distribution (ID) and out-of-distribution (OOD) performance of these estimators. Among the function families considered, **Sigmoid** performs competitively (normalization details in Section D), matching the more flexible **I-spline** in ID pinball loss and achieving better OOD calibration. Given its strong generalization and simplicity, we use **Sigmoid** as the default boundary class in the remainder of the paper.

➠ **finding 1.** *Sigmoid scaling of capability boundaries. Post-training capability boundaries are well-approximated by sigmoid functions of log-compute. The sigmoid estimator matches the more flexible I-spline in ID pinball loss and achieves the best OOD calibration error across all estimators.*

### 3.1.2. TEMPORAL STABILITY OF THE SIGMOID CAPABILITY BOUNDARY

We summarize cross-temporal transfer using two diagnostics from Section 2.3: (i) signed coverage error $(\hat{\tau} - \tau)$ and (ii) out-of-distribution pinball loss $\rho_\tau$. Negative coverage error indicates under-coverage (newer models exceed the predicted $\tau$-boundary more than intended), while positive values indicate over-coverage.

| | Pinball loss | | Calibration error | |
|---|---|---|---|---|
| **Estimator** | ID | OOD | ID | OOD |
| Constant | $5.35 \times 10^{-3}$ | $6.23 \times 10^{-3}$ | $4.12 \times 10^{-2}$ | $3.60 \times 10^{-2}$ |
| Binwise | $4.01 \times 10^{-3}$ | $5.00 \times 10^{-3}$ | $1.66 \times 10^{-2}$ | $2.81 \times 10^{-2}$ |
| I-spline | $4.00 \times 10^{-3}$ | $4.92 \times 10^{-3}$ | $1.83 \times 10^{-2}$ | $2.41 \times 10^{-2}$ |
| Sigmoid | $4.08 \times 10^{-3}$ | $4.93 \times 10^{-3}$ | $1.84 \times 10^{-2}$ | $2.21 \times 10^{-2}$ |

*Table 2.* Results averaged over three rolling splits $t = 1, 2, 3$ and six tasks. Values are absolute pinball loss and calibration error.

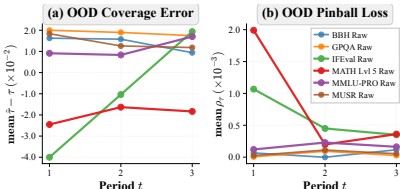

*Figure 2.* **Temporal drift and the stability of knowledge-intensive capabilities.** Left: coverage error $\hat{\tau} - \tau$. Right: pinball loss $\rho_\tau$. Both fit on $\mathcal{P}_t$ and evaluate on $\mathcal{P}_{t+1}$.

Figure 2 shows that for BBH, GPQA, MMLU-PRO, and MUSR, both diagnostics remain stable across periods (coverage error within $\pm 2\%$), indicating that a compute-only **Sigmoid** boundary transfers reliably to the next generation of open-weight models. The main departures occur on MATH LVL 5 (and to a lesser extent IFEVAL), where

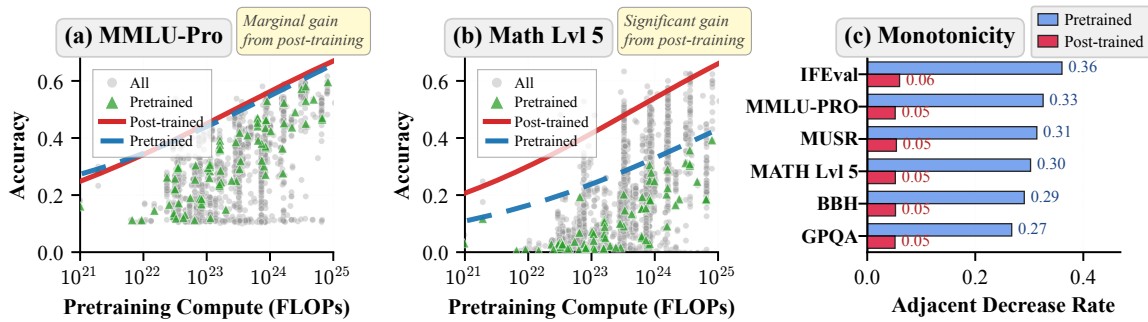

*Figure 3.* **Pre-training vs. post-training scaling laws.** Panels (a) and (b) compare capability boundaries for pretrained and post-trained models. Panel (c) compares how frequently pretrained accuracies and post-trained capability boundaries violate monotonicity in compute.

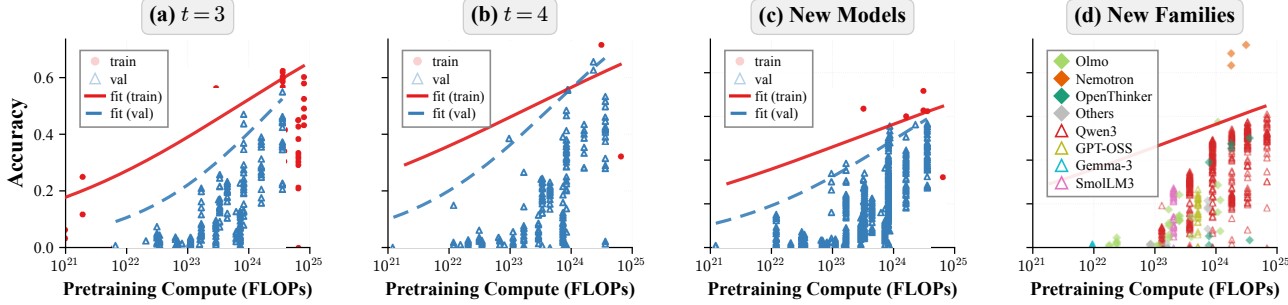

*Figure 4.* **MATH LVL 5: evaluation on newly released open-weight models.** (a) and (b): fitted sigmoid capability boundaries on leaderboard models (red) and newly evaluated models (blue) in periods $\mathcal{P}_t$ for $t \in \{3, 4\}$. (c) and (d): on PROTEUS-2K, fitted capability boundary on leaderboard models in period $\mathcal{P}_4$ (red) and on models released after the retirement of the Open LLM Leaderboard. (c) contains models from old base model families (*i.e.,* base models that already exist in the leaderboard), while (d) contains new model families.

we observe under-coverage and elevated $\rho_\tau$ on the earliest split, consistent with non-stationarity of the effective boundary over time. Bin-wise breakdowns underlying these aggregates are deferred to Section D.2.

*Remark* 3.1. A qualitatively new recipe class or architecture family could shift the attainable boundary upward at fixed compute. Under our framework this would manifest as systematic under-coverage in the next-period evaluation, motivating an updated boundary fit.

### 3.2. From Pre-training Scaling Laws to Post-training Capability Boundaries

Pre-training scaling laws (Kaplan et al., 2020; Hoffmann et al., 2022) relate training compute to model quality (e.g., benchmark accuracy) under controlled training recipes. In practice, deployed systems are almost always *post-trained* (instruction tuning, preference learning, domain adaptation), and the observed benchmark landscape reflects heterogeneous post-training pipelines and evaluation artifacts (Ouyang et al., 2022). Our $\tau$-capability boundary instead estimates a high-quantile level of performance that is *attainable after post-training* among models built on a given amount of compute.

To connect this view back to the pre-training scaling-law

perspective, we compare the *official pretrained* models (i.e., non-instruction-tuned base models) against the post-trained $\tau$-boundary in Figure 3. Two takeaways emerge.

➡ **finding 2.** *The pretrain–post-train gap is task dependent. On knowledge-intensive benchmarks (e.g.,* MMLU-PRO*), pretrained models already lie close to the post-trained boundary, whereas on reasoning and instruction-following tasks (e.g.,* MATH LVL 5*,* IFEVAL*) they sit substantially below it.*

From Figure 3 (c) we can see that among pretrained models, performance is more often *locally non-monotone* in compute: larger-compute base models can score lower than smaller ones. In contrast, for *post-trained* models the attainable upper envelope is much more consistently monotone in compute, aligning with our monotone boundary fits.

➡ **finding 3.** *Compute predicts potential more reliably than raw pretrained accuracy. Post-trained capability boundaries are consistently monotone in compute, whereas pretrained accuracies frequently violate monotonicity across model families.*

Section C provides additional results comparing pretrained and post-trained models' downstream performance.

*Remark* 3.2. Motivated by recent findings that multi-

benchmark performance is often well-approximated by low-dimensional latent capability factors (Ruan et al., 2024; Jin et al., 2025), we also run our **Sigmoid** $\tau$-capability-boundary estimator on PCA-derived factors. In our setting, the top three PCs explain $\approx 95\%$ of the total variance. Surprisingly, only the first exhibits a clear monotone increase with pre-training compute, suggesting that compute-driven progress concentrates primarily along a single dominant latent axis. Details can be found in Section H.3.

### 3.3. External Validity: Newly Evaluated Models on PROTEUS-2K

The Open LLM Leaderboard is not exhaustive: many models are never added, and new releases arrive after the leaderboard's retirement (2025-03-13). To examine external validity beyond this curated subset, we evaluate an additional 2.4k open-weight models that are absent from the leaderboard, spanning April 2024 to March 2026. Across tasks, the leaderboard-fitted boundary continues to upper-bound the best observed performance in this held-out set, with the main exceptions occurring for MATH LVL 5, consistent with the temporal non-stationarity highlighted earlier.

As shown in Figure 4, on MATH LVL 5, the capability boundary advances primarily at the high-compute end, with little evidence of systematic uplift at smaller compute. Section F reports results for the remaining tasks.

## 4. Capability Boundary Estimation under Limited Budget

### 4.1. Budget-Constrained Balanced I-Optimal Design

Evaluating *every* model across *all* tasks would give the most accurate estimate of the performance boundary, but is often prohibitively expensive. We therefore study how to select only a subset of models to evaluate under a hard evaluation budget, while still reliably recovering a well-calibrated boundary. In this section, we introduce an efficient approach to achieve this, motivated by the optimal experimental design literature (de Aguiar et al., 1995; Goos & Jones, 2011; Goos et al., 2016; Smucker et al., 2018). The I-optimal design allocates evaluations to minimize the average predictive variance of the estimated capability boundary across compute regimes (Pukelsheim, 2006). Intuitively, it concentrates budget on the most informative models so the fitted sigmoid frontier has uniformly low uncertainty rather than low error at a few points.

**Cost and budget.** For each model $i$, let $c_i$ be its parameter count. We assume evaluation cost grows roughly linearly with model size. Let $\mathcal{P}_t$ denote the set of candidate models in training period $t$ and $C_t = \sum_{i \in \mathcal{P}_t} c_i$ the total cost of evaluating all of them. Given a user-chosen $\alpha \in [0, 100]$,

we allocate a per-period budget $U_t = \frac{\alpha}{100} C_t$, and seek a subset $S_t \subseteq \mathcal{P}_t$ with $\sum_{i \in S_t} c_i \leq U_t$ that yields accurate OOD predictions for the next period.

**Sigmoid boundary and information matrix.** Recall that the $\tau$-quantile performance boundary is modeled as a sigmoid function of log-compute $z = \log_{10}(\texttt{FLOPs})$,

$$q_\tau(z; \theta) = y_0 + L\,\sigma(a + bz),$$

with parameters $\theta = (y_0, L, a, b)$ and $\sigma(t) = 1/(1 + e^{-t})$. Let $j(z; \theta) = \begin{bmatrix} 1, & \sigma, & L\sigma(1-\sigma), & L\sigma(1-\sigma)z \end{bmatrix}^\top$ denote the Jacobian of $q_\tau$ with respect to $\theta$ as a column vector, where we write $\sigma = \sigma(a + bz)$ for brevity, and we evaluate it at a nominal parameter $\theta_0$ obtained from an initial estimate. For a selected set $S$ we define the local information matrix

$$M(S) = \sum_{i \in S} j(z_i; \theta_0) j(z_i; \theta_0)^\top.$$

For numerical stability, we use a ridge prior $M_0 = \eta I$ with $\eta = 10^{-9}$, and let $\Sigma_\theta(S) \approx (M_0 + M(S))^{-1}$.

We consider the bin partition introduced in Section 2.2 and let $\tilde{z}_b$ denote the midpoint of bin $b$. The delta method gives an approximate predictive variance of the boundary at $\tilde{z}_b$,

$$v_b(S) \approx j(\tilde{z}_b; \theta_0)^\top \Sigma_\theta(S)\, j(\tilde{z}_b; \theta_0).$$

We then define the I-optimal predictive-variance objective

$$\Phi_{\text{info}}(S) = -\sum_{b=1}^{B} w_b\, v_b(S), \tag{1}$$

where we use uniform bin weights $w_b = 1/B$. In other words, $\Phi_{\text{info}}$ characterizes the *average* predictive variance across bin midpoints. Maximizing $\Phi_{\text{info}}$ is therefore equivalent to minimizing this average predictive variance.

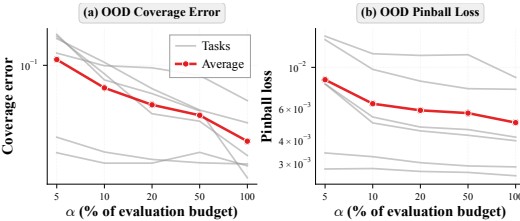

*Figure 5.* Performance of balanced I-optimal design as a function of budget parameter $\alpha$, averaged over $t = 1, 2, 3$.

**Bin-balanced design.** To explicitly encourage coverage of all compute regimes, we augment the objective with a *bin-balance* term. Let $b(i) \in \{1, \ldots, B\}$ be the bin index of model $i$ and $n_b(S) = \big|\{i \in S : b(i) = b\}\big|$ denote the number of selected models in bin $b$. We define

$$\Phi_{\text{bal}}(S) = \sum_{b=1}^{B} \log\big(n_b(S) + \varepsilon\big), \tag{2}$$

where $\varepsilon > 0$ is a small constant. This imposes a preference for designs that distribute models more evenly across bins.

**Balanced I-optimal objective.** Our final design criterion for period $t$ combines the predictive-variance and bin-balance terms:

$$\Phi_\lambda(S_t) = \Phi_{\text{info}}(S_t) + \lambda\,\Phi_{\text{bal}}(S_t), \quad \text{s.t.} \sum_{i \in S_t} c_i \leq U_t,$$

where $\lambda \geq 0$ trades off boundary uncertainty against bin coverage. Setting $\lambda = 0$ recovers standard I-optimality. We approximately maximize $\Phi_\lambda(\cdot)$ under the budget constraint using a standard greedy gain-per-cost heuristic over models in $\mathcal{P}_t$. This procedure only uses model metadata $(z_i, c_i)$ and the local Jacobian. Details can be found in Section J.

**Empirical behavior.** Figure 5 shows average OOD performance on period $t + 1$ for $t \in \{1, 2, 3\}$ as a function of the budget parameter $\alpha$. Across periods and tasks, error decreases rapidly as $\alpha$ increases and stabilizes between 20% and 50%. In particular, for GPQA and MUSR we obtain near-identical boundary estimates to the full-data fit using as little as $\alpha = 5\%$ of the evaluation budget.

➠ **finding 4.** *Efficient prescriptive scaling. The balanced I-optimal design recovers near-full-data capability boundaries using $\approx 20\%$ of evaluation budget, and as little as 5% on tasks such as GPQA and MUSR.*

# 5. Case Studies: Saturation and Contamination Diagnostics

Up to now, our prescriptive scaling framework has focused on estimating capability boundaries as a function of pre-training compute. In modern evaluation pipelines, however, two additional issues are central: (i) task-dependent saturation narratives on public leaderboards, where the relationship between scale and scores evolves over time, and (ii) time-dependent evaluation artifacts, including contamination and *training on the test task* (Dominguez-Olmedo et al., 2024). In this section, we show how capability-boundary estimation yields quantitative diagnostics for both, and we additionally use frontier-model leaderboards to probe external validity beyond open-weight models.

## 5.1. Task-dependent Saturation on Leaderboards

Benchmark saturation—when static test sets lose headroom and discriminative power as models improve—is a recurring theme in evaluation: rapid ceiling effects on earlier leaderboards have motivated harder successor suites (e.g., SuperGLUE after GLUE) (Wang et al., 2019), while dynamic benchmarking frameworks explicitly anticipate and mitigate saturation by iteratively refreshing evaluation data (Kiela et al., 2021). At the ecosystem level, large-scale analyses show that near-saturation emerges quickly for many benchmarks across vision and NLP, suggesting that tracking *how* the attainable envelope evolves can be as important as reporting absolute scores (Ott et al., 2022). This pressure

has spurred "reset" benchmarks intended to restore headroom (e.g., MMLU-Pro (Wang et al., 2024) and Humanity's Last Exam (Phan et al., 2025)) as well as rolling evaluations designed to remain challenging under rapid capability growth (White et al., 2024).

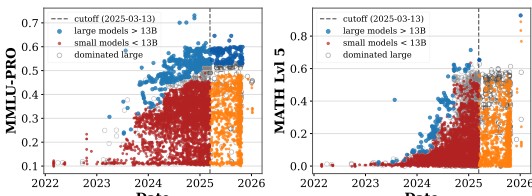

*(a)* **MMLU-Pro (knowl-** *(b)* **MATH Lvl 5 (reason-**
**edge):** weaker saturation. **ing):** stronger saturation.

*Figure 6.* **Task-dependent saturation on Open LLM Leaderboard v2.** "Dominated large" marks large ($> 13B$) models whose task score is below the cumulative best score achieved by small models up to that date (the small-model boundary). MMLU-Pro appears less saturated than MATH Lvl 5.

In this subsection, we revisit the saturation problem with a focus on its dependency on model size. Concretely, we are interested in the following question: *does saturation occur simply because people are building larger models?* Recent work argues that the relationship between scale and capability is becoming less predictable, and that "bigger" does not reliably imply "better" on open benchmarks (Hooker, 2025). Our replication on Open LLM Leaderboard v2 produces a quantitative, *task-dependent* account of this phenomenon.

Figure 6 contrasts a mostly knowledge-based benchmark with a pure reasoning benchmark. These saturation diagnostics are parameter-count centric and are intended as a complementary lens to our compute-based boundaries: they summarize *whether small models quickly reach their attainable boundary on a benchmark and whether larger models retain a persistent advantage*.

### 5.1.1. QUANTIFYING SATURATION WITH A SIZE–TIME BOUNDARY MODEL

To quantify how *model size* shifts the attainable performance on a benchmark while accounting for time-dependent effects, we fit a simple size–time boundary model. For benchmark $b$, let $q_\tau^b(x, t) := Q_\tau(Y_b \mid X = x, T = t)$ denote the attainable $\tau$-quantile score at size $x$ and time $t$. We model $\text{logit}(q_\tau^b(x, t))$ as

$$\text{logit}\big(q_\tau^b(x_i, t_i)\big) = \alpha_b + \beta_b x_i + \phi_b(t_i) + \delta_b g_b(t_i) + \theta_b\, x_i g_b(t_i),$$

where $y_{ib} \in [0, 1]$ is model $i$'s score on benchmark $b$, $x_i = \log(\#\text{params}_i)$, and $t_i$ is release time. Here $\phi_b(t)$ captures a smooth time trend, and $g_b(t) \in \{0, 1\}$ is a late-period indicator (we use the cutoff date in Figure 6). We fix $\tau = 0.98$. Intuitively, the quantity $\beta_b + \theta_b g_b(t)$ can be viewed as the marginal size effect on the capability boundary at time $t$,

while $\hat{q}_\tau(13\text{B}, g{=}1)$ characterizes the late-period attainable boundary for "small" (13B-parameter) models.

We fit the model on Open LLM Leaderboard data together with our newer-model evaluations. Comparing MATH LVL 5 with MMLU-PRO (Table 3), we find that the estimated 13B attainable boundary is much higher on Math in the late period ($\hat{q}_{0.98} \approx 0.94$) than on MMLU-Pro ($\approx 0.52$), consistent with small models approaching the top boundary on Math but remaining substantially below the boundary on MMLU-Pro, where larger models retain dominance.

| Benchmark | $\hat{\beta}$ | $\hat{\beta} + \hat{\theta}$ | $\hat{q}_{0.98}(13\text{B}, g{=}0)$ | $\hat{q}_{0.98}(13\text{B}, g{=}1)$ |
|---|---|---|---|---|
| MATH Lvl 5 | 0.25 | 0.76 | 0.03 | 0.94 |
| MMLU-Pro | 0.00 | 0.47 | 0.15 | 0.52 |

*Table 3.* Key fitted statistics from the size–time boundary model. Here $g{=}0$ and $g{=}1$ denote the earliest and latest times in the pooled dataset, respectively; $\hat{\beta} + \hat{\theta}$ is the late-period marginal size effect on the boundary.

➠ **finding 5.** *Task-dependent small-model ceilings. Knowledge-intensive capability, such as MMLU-Pro, remains scale-limited in current practice; post-training does not eliminate the advantage of larger base models. For instance, 13B models reach $\hat{q}_{0.98} \approx 0.94$ on MATH Lvl 5 but only $\approx 0.52$ on MMLU-Pro in the latest period.*

## 5.2. Contamination or Train-on-test-task Diagnostics on Frontier Benchmarks

Beyond open-weight leaderboards, we use frontier-model evaluations to probe external validity and to construct contamination-oriented diagnostics. We first test whether the sigmoid boundary remains competitive relative to the more flexible I-spline class on closed-source frontier models with known compute. We then examine a simple cross-benchmark shift test designed to detect post-release score inflation consistent with benchmark-specific contamination.

### 5.2.1. FRONTIER-MODEL SCALING ON GPQA

Epoch AI evaluates many closed-source frontier models. We fit the capability boundary of GPQA diamond (Rein et al., 2024) based on models with known compute. As in Section 3.1.1, we compare the sigmoid estimator with the more complex I-spline estimator in Figure 7a and find they are largely similar, supporting the external validity of the sigmoid scaling law on frontier models. Additional results for boundaries fitted from other publicly available frontier leaderboards are provided in Section H.1.

### 5.2.2. A CROSS-BENCHMARK SHIFT TEST

We also re-examine contamination by exploiting the hypothesis that both benchmark scores are (monotone) sigmoid functions of pretraining compute, which implies an approximately linear relationship between their logit-transformed

true accuracies. Since all models post-date MATH-500, any MATH-500 inflation can affect all points, whereas AIME-2025-specific inflation should only affect models released after Feb. 6th, 2025. This motivates the regression

$$\text{logit}(0.01 y_i) = \alpha + \beta\,\text{logit}(0.01 m_i) + \gamma\,\mathbb{1}\{\text{post-AIME}\} + \varepsilon_i, \tag{3}$$

Here $y_i$ is the AIME 2025 accuracy and $m_i$ is the corresponding MATH-500 accuracy for the same model. A positive $\gamma$ corresponds to systematically higher AIME 2025 performance than would be predicted from MATH-500.

Restricting to the overlapping range of MATH-500 scores across the two release groups ($n = 90$), we estimate a positive but not statistically significant shift (p-value $= 0.15$). In other words, we find no clear aggregate evidence that post-release AIME-2025 scores are unusually high relative to what MATH-500 performance would predict, though modest contamination effects cannot be ruled out.

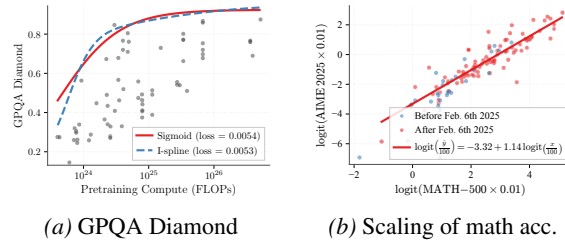

*(a)* GPQA Diamond     *(b)* Scaling of math acc.

*Figure 7.* Scaling laws for frontier models.

## 6. Related Works

**Scaling laws and downstream predictability.** Classical scaling laws relate model size, data, and compute to pretraining loss under controlled settings (Kaplan et al., 2020; Hoffmann et al., 2022). Translating these forecasts into actionable guidance for future model development (McCandlish et al., 2018; Hernandez et al., 2021; Kaplan et al., 2020; Zhang et al., 2025c), however, remains an open challenge. Particularly, their extension to downstream task performance has proven substantially noisier and less reliable (Chen et al., 2024; Lourie et al., 2025; Schaeffer et al., 2024). Recent work on observational scaling laws studies compute–performance relationships using heterogeneous trained models (Ruan et al., 2024), and shows that post-training on test or target tasks can make downstream performance appear more predictable, while also introducing confounding temporal effects (Dominguez-Olmedo et al., 2024). We build on this line by focusing on attainable performance boundaries rather than average trends.

**Downstream performance forecasting and scaling analyses.** Under controlled regimes, downstream accuracy often follows simple power laws in training compute, enabling reliable FLOP-to-accuracy extrapolation (Krajewski et al., 2025). In realistic settings, however, emergent abilities, metric noise, and heterogeneous pipelines frequently violate

these trends (Schaeffer et al., 2024). To address this, prior work proposes two-stage regressions from compute to pre-training loss to downstream metrics (Chen et al., 2024), clustering methods that isolate predictable task subsets (Xu et al., 2025), and rectified scaling laws that forecast fine-tuning performance via a learned data-size term (Lin et al., 2024). Large-scale observational analyses from Epoch AI further inform forecasting by tracking compute, data, and performance trajectories across thousands of models (Epoch AI, 2022). In contrast to fine-tuning–centric or point-forecasting approaches, we study post-training capability boundaries, estimating high-quantile, prescriptive performance boundaries from heterogeneous models and analyzing their temporal stability and data efficiency.

**Post-training and performance variance.** Post-training techniques such as instruction tuning and preference optimization can substantially shift downstream performance without changing pretraining compute (Ziegler et al., 2019; Ouyang et al., 2022). Recent studies emphasize that post-training often elicits latent capabilities, leading to large variance among models trained with similar FLOPs (Donoway et al., 2025; Jin et al., 2025). In parallel, targeted pretraining methods demonstrate that aligning pretraining data with downstream objectives can yield significant gains (Brandfonbrener et al., 2024; Mizrahi et al., 2025). Our work complements these algorithmic advances by abstracting over heterogeneous post-training pipelines and treating models with similar pretraining FLOPs as a single domain, enabling large-scale observational analysis that is better aligned with engineering decision-making.

## 7. Conclusion

We introduced prescriptive scaling, a decision-oriented framework for mapping pre-training compute budgets to reliable, high-probability downstream performance expectations under contemporary post-training practice. By estimating high-quantile capability boundaries from large, heterogeneous model populations, we show that attainable post-training performance is well-approximated by simple, monotone sigmoid functions of log-compute and is temporally stable for most tasks. Notable exceptions, such as math reasoning, exhibit a shifting boundary, highlighting where algorithmic progress continues to move the attainable envelope. Beyond forecasting, our approach enables practical diagnostics for saturation, contamination, and evaluation efficiency, allowing recovery of near-full boundaries with a small fraction of evaluation budget. Together, these results position capability boundaries as a practical tool for budgeting, monitoring, and interpreting progress in language models as scaling regimes evolve.

## Acknowledgment

We thank Moritz Hardt for the discussion of evaluation and benchmarking, and the open-source community for realeasing artifacts through Huggingface. VS is partially supported by the Schmidt Sciences' Trustworthy AI Program award on AI Safety in the Inference-time Compute Paradigm; SK acknowledges the support from the National Science Foundation Grant under award IIS 2229881; HZ and SK acknowledge the Chan Zuckerberg Initiative Foundation for establishing the Kempner Institute for the Study of Natural and Artificial Intelligence.

## Impact Statement

This paper presents work whose goal is to advance the field of LLM evaluation. There are many potential societal consequences of our work, none of which we feel must be specifically highlighted here.

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

# Appendices

## Table of Contents

# A. Pinball Loss and Quantile Regression

## A.1. Properties of Pinball Loss

Our sigmoid boundaries are not fitted with a symmetric squared loss, but with a *high-quantile pinball loss* (Koenker & Bassett, 1978) that explicitly targets the upper envelope of the data. For a residual

$$r = y - q_\tau(z; \theta),$$

the true pinball loss at quantile level $\tau \in (0, 1)$ is

$$\rho_\tau(r) = \max(\tau r, \ (\tau - 1)r),$$

which is piecewise linear with a kink at $r = 0$. Minimizing its expected value recovers the $\tau$-quantile of the (conditional) response distribution (Koenker & Bassett, 1978). In practice we use a smoothed variant

$$\tilde{\rho}_\tau(r) = \frac{1}{\kappa} \log(1 + e^{\kappa r}) + (\tau - 1)\, r,$$

with $\kappa \approx 50$ (Figure 8a), which is numerically stable yet visually indistinguishable from the sharp pinball loss away from $r = 0$.

The key property of the pinball loss is its *asymmetry*. As shown in Figure 8b, the gradient

$$g_\tau(r) = \frac{\partial \tilde{\rho}_\tau(r)}{\partial r}$$

is approximately $\tau - 1$ for $r < 0$ and $\tau$ for $r > 0$. For high quantiles (e.g., $\tau = 0.98$), this means that points lying *above* the boundary (positive residuals, $y > q_\tau$) incur a much larger gradient magnitude than those below it. Intuitively, the optimizer is heavily penalized whenever the boundary lies *below* high-performing models, but it almost ignores points that underperform relative to the current boundary. This is precisely the behavior we want when estimating a capability boundary: the boundary should track the best models at a given compute budget, not the mean.

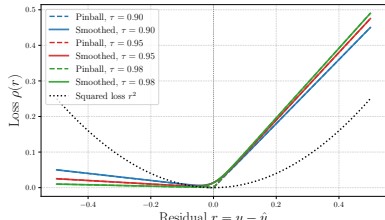
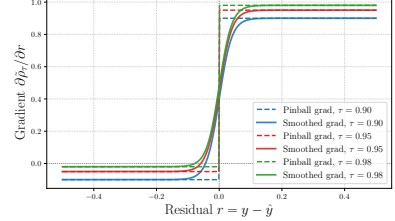
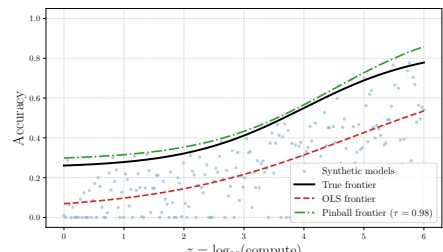

*(a)* Smoothed vs. true pinball loss and squared loss as a function of residual $r = y - \hat{y}$.

*(b)* Gradients of the pinball loss, highlighting asymmetric weighting of over/under prediction.

*(c)* True boundary vs. squared-loss and pinball-loss fits on synthetic scaling-law data.

*Figure 8.* Visualizing the pinball loss and its effect on boundary fitting. **(a)** The true pinball loss and its smoothed approximation for several quantile levels $\tau$; smoothing (softplus) affects only a narrow band around $r = 0$. **(b)** The corresponding gradients, showing that positive residuals (boundary below data) produce much larger gradients than negative residuals. **(c)** On synthetic data, the pinball boundary closely tracks the upper envelope, whereas the squared-loss boundary is pulled toward the bulk of underperforming points.

Figure 8c illustrates the practical impact of this choice on a synthetic scaling-law dataset. The black curve denotes the true boundary used to generate the data; most points are sampled below it, with a few small above-boundary outliers. Fitting a sigmoid with a symmetric squared loss pulls the boundary toward the bulk of the cloud, underestimating the achievable performance at high compute. In contrast, fitting with the smoothed pinball loss at $\tau = 0.98$ yields a boundary that closely matches the true upper envelope. Together, Figures 8a–8c show that the pinball loss provides a principled and computationally convenient way to implement a high-quantile, envelope-seeking objective for capability boundaries.

## A.2. Performance Frontier via Quantile Regression

A natural question is: if we are interested in a "capability boundary," why not regress on the maximal observed score at each compute level rather than on a high conditional quantile? In this subsection, we explain why we instead target a high quantile—implemented via the pinball loss—and why we view this as the more meaningful object for our purposes.

- In our setting, the number of models per compute bin is highly uneven and changes over time. The sample maximum in a bin is an extremely high-variance statistic whose expectation increases with the number of draws, even if the underlying distribution of model performance at that compute were fixed. A regression on bin-wise maxima would therefore confound two effects: genuine changes in the performance distribution as compute increases, and incidental variation in how many models happened to be trained, how hard different groups searched hyperparameters, or how aggressively they pruned weak runs. In contrast, a conditional quantile

$$q_\tau(z) \quad \text{with} \quad \tau \in (0,1)$$

is defined in terms of the underlying distribution of scores at compute level $z$, and does not explode just because one period happens to contain more models in a particular bin. High-quantile regression with the pinball loss is a standard, well-understood estimator in this regime, and empirically leads to much smoother, more stable boundarys than directly regressing on bin-wise maxima.

- What we ultimately want is not the performance of the single luckiest model ever trained at a given compute value, but a statement of the form: if we train a model at compute $C$ using contemporary practices, what level of performance can we expect to achieve with high probability? A high conditional quantile has a direct probabilistic semantics that matches this question. If $Z = \log_{10} C$ and $Y$ is the score on a given task, then an ideal quantile boundary $q_\tau$ satisfies

$$\Pr\big(Y \leq q_\tau(Z) \,\big|\, Z \in b\big) = \tau$$

for each compute bin $b$. This is exactly the coverage property we evaluate in Section 2: we can check, bin by bin and period by period, whether the fitted boundary over- or under-covers the empirical distribution of models. There is no analogous notion of coverage for a regression on maxima; one would simply be fitting a smooth curve through extreme points, without a built-in way to say what fraction of future models should lie above or below it.

- In realistic training processes there are occasional outlier models: unusually strong models that benefited from, for instance, severe data contamination. The maximum is by construction driven entirely by such extremes. A high, but not extreme, quantile such as $\tau \approx 0.98$ behaves differently. In bins with a reasonable number of models, the 98th percentile is already "near the top" of what the given compute range could produce, but it is much less sensitive to a single outlier run. In our experiments, this choice yields boundaries whose in-sample and out-of-sample coverage errors are on the order of $1\% - 2\%$ across tasks and periods, and whose behavior is stable under modest changes in binning and regularization.

- Finally, there is a conceptual point about what we mean by a "boundary". In practice, one rarely cares about the single best model that anyone, anywhere, has ever managed to train at a given compute value. What matters is the level that is reliably achievable by following a reasonably competitive pipeline, perhaps with some hyperparameter tuning but without assuming arbitrarily many parallel bets. A high-quantile boundary formalizes this notion of *reliably reachable top-tier performance*: it separates out the typical high end of the distribution from the one-off lucky run. For these reasons we treat $q_\tau(\cdot)$, rather than the conditional maximum, as the primary object of interest, and we use the pinball loss to estimate it in a way that can be directly calibrated and validated. This interpretation is relative to the population of reported models we observe; substantial selection effects could shift the implied boundary.

## B. Additional Details for Section 2.1–2.3

### B.1. Concrete Illustrative Outlier Example

As a concrete example, the model `qingy2024/Benchmaxx-Llama-3.2-1B-Instruct` achieves 0.83 and 0.48 raw accuracies on BBH and MATH Lvl 5, respectively, while the second-best models using `Llama-3.2-1B` as their base only reach 0.36 and 0.08. Empirically, broadly applicable algorithmic improvements are typically reproduced by multiple independent derivatives (e.g., fine-tuning variants), whereas isolated spikes can reflect idiosyncratic effects such as benchmark-specific overfitting or leakage. We include this example solely to motivate the use of high-quantile estimation (which is more robust than conditional maxima); we do not attempt to verify or attribute the underlying cause.

### B.2. Full Bin Construction Algorithm for the Binwise Model

We use group-aware equal-mass binning on the training $z$-values. Let $N$ denote the number of training samples, and let the sorted training values be grouped into unique levels with counts $\{(u_g, n_g)\}_{g=1}^G$ so identical $z$ values are never split

across bins. Given a target bin count $B$ and a minimum bin size $m_{\min}$, we set $B_{\mathrm{eff}} = \min(B, G)$ and a target mass $m = \max(m_{\min}, \lceil N/B_{\mathrm{eff}} \rceil)$. We sweep the unique levels in increasing order, accumulating counts until the running total reaches $m$ (or the last group), then place a bin boundary at that unique value. If any resulting bin has fewer than $m_{\min}$ samples, we iteratively merge it with an adjacent bin by removing a boundary (merging with the smaller neighbor when there is a choice) until all bins satisfy the minimum size. This yields edges $e_0 < \cdots < e_{B'}$ (with $B'$ the final number of bins after merging), which define bins $[e_{b-1}, e_b]$ used for both training and evaluation.

### B.3. Full I-spline Definition

**I-spline estimator.** We use an I-spline basis to parameterize a flexible monotone function of $z$ (Ramsay, 1988). Let $\{\kappa_\ell\}_{\ell=0}^K$ be a nondecreasing knot sequence and let $\{M_j(\cdot)\}_{j=1}^J$ denote the associated nonnegative M-spline basis (order $p$). Define I-spline basis functions

$$I_j(z) = \int_{\kappa_0}^z M_j(u)\, du, \tag{4}$$

so each $I_j(z)$ is nondecreasing in $z$. We then model

$$g(z) = a_0 + \sum_{j=1}^J w_j I_j(z), \qquad w_j \geq 0, \tag{5}$$

which guarantees that $g(z)$ is nondecreasing. Predictions are $q_\tau^{\mathrm{I}}(z) = \sigma(g(z))$, ensuring a monotone saturating boundary in $[0, 1]$.

## C. Pre-training vs. Post-training Diagnostics

In this appendix, we provide additional details for the comparison between pretrained checkpoints and post-trained variants in Section 3.2.

**Pretrained subset and evaluation protocol.** We leverage the "official" and "pretrained" labels available from the Open LLM Leaderboard to construct the set of official pretrained checkpoints. This choice is closer to existing scaling-law studies (Ruan et al., 2024; Dominguez-Olmedo et al., 2024), which are mostly based on popular open-weight pretrained model families such as Llama, Qwen, and Gemma.

**Quantities reported.** Let $q_\tau^{\mathrm{post}}(z)$ denote the fitted post-trained sigmoid capability boundary. We summarize the relationship between pretrained checkpoints and post-training capability using:

- **Gap-to-boundary:** $q_\tau^{\mathrm{post}}(z_i) - y_i^{\mathrm{pre}}$ for each pretrained checkpoint $i$ (how far the base checkpoint is below the post-training envelope at the same compute).

- **Post-training lift (paired when possible):** $y_i^{\mathrm{post\text{-}best}} - y_i^{\mathrm{pre}}$, where $y_i^{\mathrm{post\text{-}best}}$ is the best post-trained score among checkpoints sharing the same base model identity/compute as $i$.

Figure 9 visualizes these quantities across all six tasks. The results indicate that post-training provides the largest gains on IFEVAL and MATH LVL 5, with much smaller gains on MMLU-PRO, BBH, GPQA, and MUSR; this qualitative pattern holds across the observed range of pretraining compute. Figure 10 overlays the fitted pretrained capability boundaries with the corresponding post-trained boundaries across all six tasks.

## D. Details and Additional Analyses for Section 3

### D.1. Omitted Details in Section 3

This section collects a few plotting and normalization conventions used in Section 3.

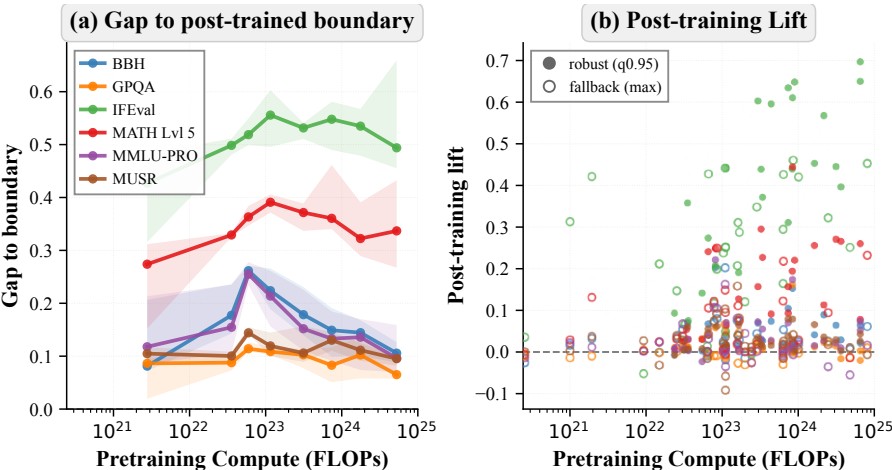

*Figure 9.* **Summary diagnostics connecting pretraining compute to post-training capability.** These plots quantify (i) how far pretrained checkpoints lie below the post-trained $\tau$-capability boundary, and (ii) how much post-training can lift performance at fixed compute.

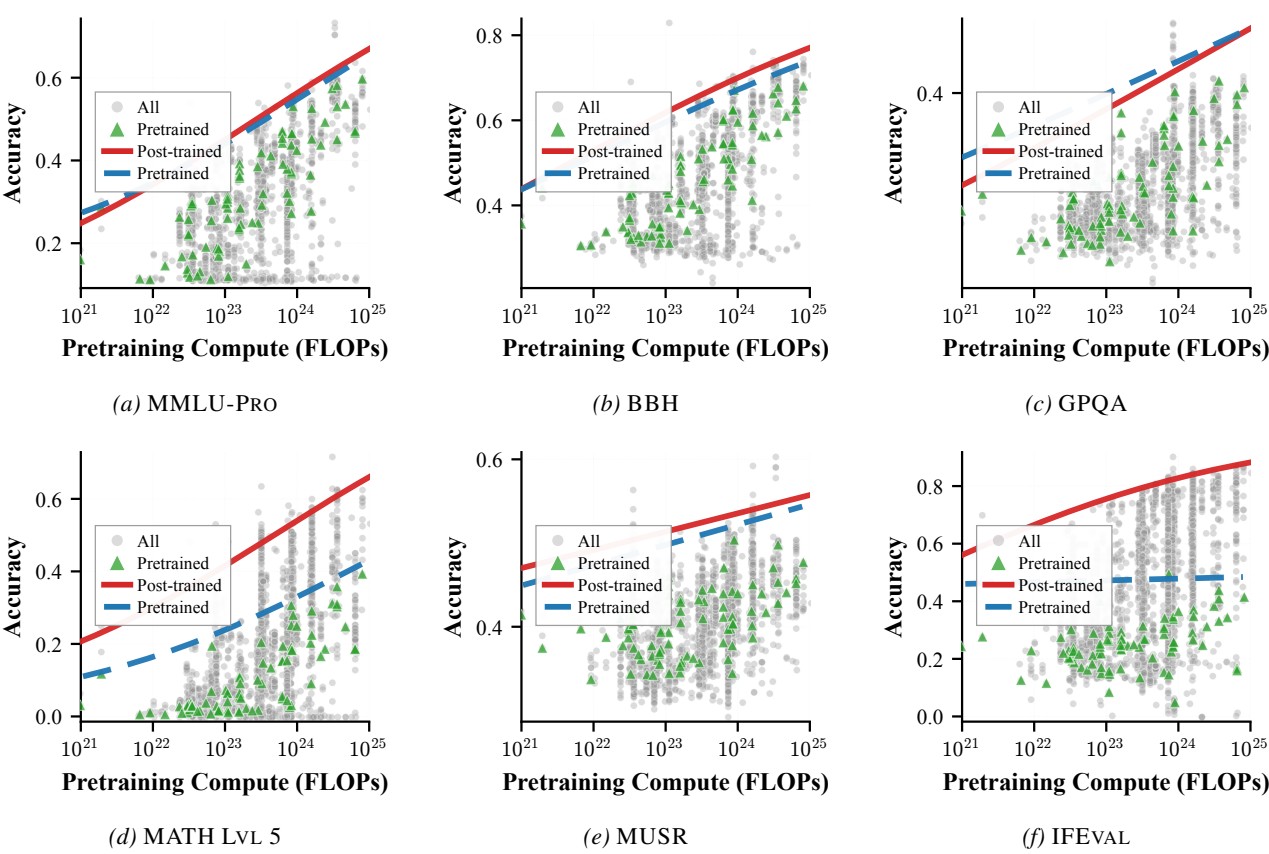

*Figure 10.* **Pretrained vs. post-trained overlays across all tasks.** See Figure 3 for the legend description.

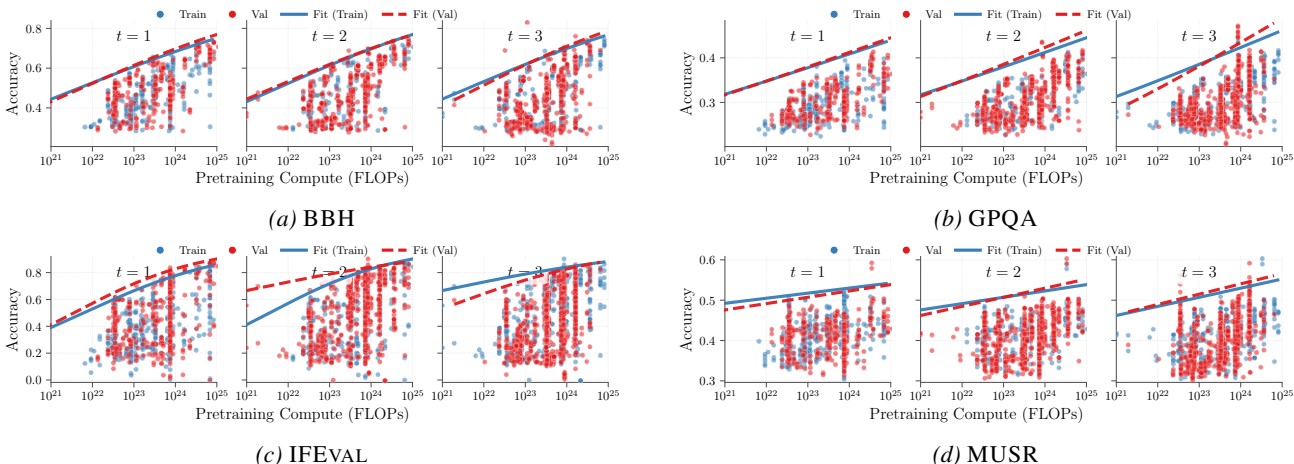

*Figure 11.* **Sigmoid capability boundaries over time.** Complementary of Figure 1.

**Rolling train/validation protocol and overlap restriction.** For each temporal split $t \in \{1, 2, 3\}$, we fit each boundary estimator on $\mathcal{P}_t$ and evaluate out-of-distribution (OOD) on $\mathcal{P}_{t+1}$. To avoid extrapolating beyond observed compute, OOD evaluation is restricted to the overlap of the training and validation ranges in $z = \log_{10} C$.

The table below provides detailed base model information for models in all four time periods. We only include base models that are used at least ten times. One can see that it covers almost all mainstream base models during that time period.

| Base model | ≤2024-06 | 2024-07..2024-09 | 2024-10..2024-12 | 2025-01..2025-03 |
|---|---|---|---|---|
| gemma-1-2b | 13 | – | – | – |
| gemma-2-27b | – | – | 14 | – |
| gemma-2-2b | – | 13 | 12 | – |
| gemma-2-9b | – | 27 | 65 | 21 |
| llama-2-13b | 15 | – | – | – |
| llama-2-70b | 14 | – | – | 10 |
| llama-2-7b | 24 | – | 10 | 13 |
| llama-3-70b | 26 | – | – | – |
| llama-3-8b | 167 | 113 | 156 | 197 |
| llama-3.1-70b | – | 12 | 14 | – |
| llama-3.1-8b | – | 89 | 65 | 50 |
| llama-3.2-1b | – | – | 22 | 25 |
| llama-3.2-3b | – | – | 43 | 35 |
| mistral-7b | 132 | 63 | 44 | 49 |
| phi-3-4b | 10 | – | 14 | – |
| phi-4-14b | – | – | – | 48 |
| qwen2-0.5b | – | – | 22 | 23 |
| qwen2-1.5b | – | – | – | 56 |
| qwen2-72b | 12 | 14 | – | – |
| qwen2-7b | 23 | 12 | 58 | 133 |
| qwen2.5-0.5b | – | – | 57 | 109 |
| qwen2.5-1.5b | – | – | – | 11 |
| qwen2.5-14b | – | – | 84 | 185 |
| qwen2.5-32b | – | – | 28 | – |
| qwen2.5-3b | – | – | 25 | 80 |
| qwen2.5-7b | – | 14 | 53 | 70 |

**Relative improvements in Table 2.** In Table 2, we report percent changes relative to the constant baseline **Constant**. For a metric value $m$ (pinball loss or coverage error), the plotted relative change is

$$\Delta_\% \;=\; 100 \times \frac{m_{\text{method}} - m_{\text{Constant}}}{m_{\text{Constant}}}.$$

Thus, more negative values indicate better performance than **Constant**.

**Bin-wise coverage heatmaps in Figure 14.** Coverage is evaluated using the bin-wise coverage metric from Section 2.3. Bins are constructed *on the training period only* in $z$ (never splitting identical $z$ values), and the same bin edges are reused for OOD evaluation on $\mathcal{P}_{t+1}$. Within each bin, we compute empirical coverage $\hat{\tau}$ and report the signed deviation $\hat{\tau} - \tau$. Negative values indicate *under-coverage*: more than a $(1-\tau)$ fraction of models in that bin exceed the predicted $\tau$-boundary fit on $\mathcal{P}_t$.

### D.2. Bin-wise Diagnostics underlying Figure 2

Figure 12 and 13 report the bin-wise OOD breakdowns of three benchmarks: MMLU-PRO, MATH LVL 5 and IFEVAL, that are omitted from the main paper for space. These plots use log-compute bins constructed on the training period only (see Section D) and evaluate only on the train–OOD overlap in $z$.

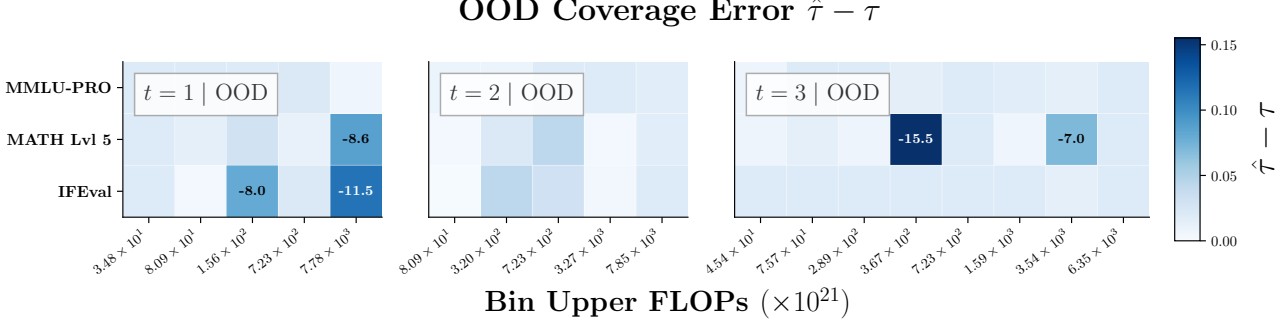

*Figure 12.* **Bin-wise coverage across periods (supplement to Figure 2 (Left)).** Bin-wise signed OOD coverage error within log-compute bins when fitting on $\mathcal{P}_t$ and validating on $\mathcal{P}_{t+1}$, for $t = 1, 2, 3$.

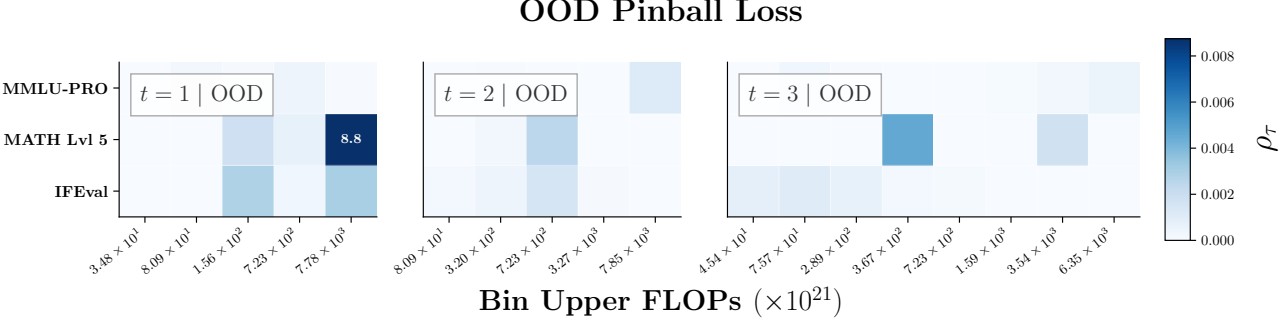

*Figure 13.* **Bin-wise pinball loss across periods (supplement to Figure 2 (Right)).** Bin-wise OOD pinball loss within log-compute bins when fitting on $\mathcal{P}_t$ and validating on $\mathcal{P}_{t+1}$, shown for MMLU-PRO, MATH LVL 5, and IFEVAL with $t = 1, 2, 3$.

**Localization of deviations.** The bin-wise heatmaps show that the largest deviations in both coverage and $\rho_\tau$ are concentrated in a small subset of mid-to-high compute bins (not uniformly across compute), rather than reflecting pervasive misfit across all scales.

The complete in-distribution (ID) and out-of-distribution (OOD) coverage error and pinball loss are visualized in Figure 14 and 15. We can see that apart from the aforementioned three benchmarks, the remaining ones all have mild ID and OOD errors across different compute bins, implying that the scaling remains stable on these benchmarks.

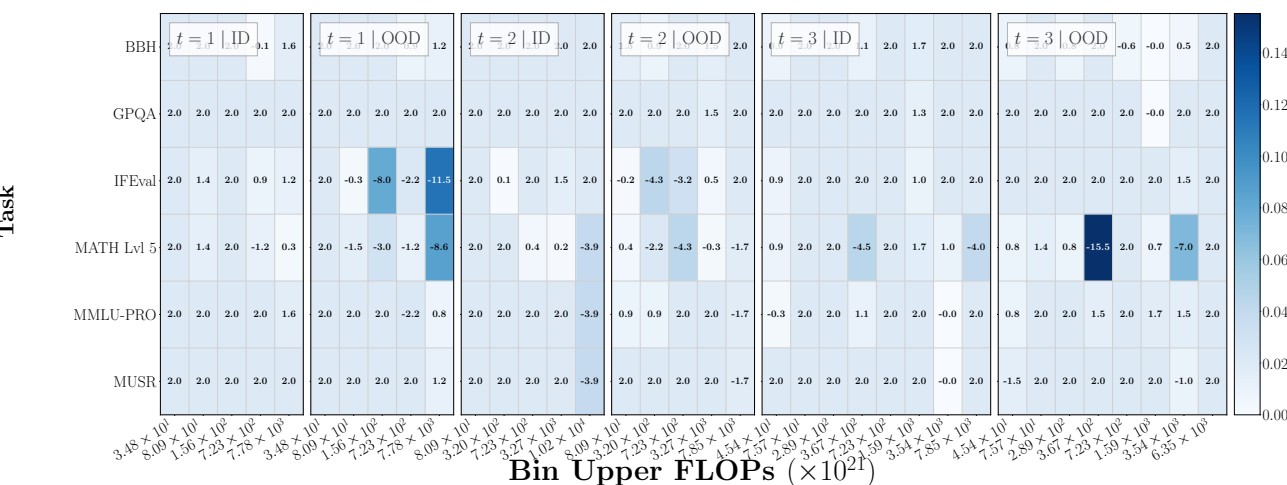

*Figure 14.* In-distribution and out-of-distribution coverage error across log-compute bins for train $(\mathcal{P}_t)$ and validation $(\mathcal{P}_{t+1})$, $t = 1, 2, 3$ across six Open LLM Leaderboard tasks.

## E. Scaling Laws for Model Size

In this section, we provide complementary results for the capability boundaries as functions of the model size. This is in contrast with classical scaling laws (Kaplan et al., 2020) use either the pretraining compute or the model size plus the pretraining token size to predict downstream task performance. Using model sizes as the single predictive factor is useful, since it informs us how much capability a small model can acquire with an unbounded amount of pretrained data.

Figure 16 compares the scaling of pretrained and post-trained models. The findings are largely similar with the pretraining-compute-based counterparts. One novel finding is that on MUSR, while larger *pretrained* models do not show a clear benefit over smaller ones, *post-trained* models have a much clearer scaling in terms of model size. This suggests that for multi-step reasoning tasks, larger models could have larger potentials than smaller ones even if the base model accuracies are similar.

In Figure 17, we further compare the temporal changes of the sigmoid capability boundaries across different tasks. While MATH LVL 5 and IFEVAL performances initially improve under fixed model size, the curves tend to stablize in later periods.

## F. Newly Evaluated models

In this section, we provide complete results for the newly evaluated open-weight models. Concretely, we evaluate the performance of two different types of models:

- Models available on huggingface with the most number of likes, filtered by compatibility with the lm-eval-harness. A list of base families of these models is given in Table 5.

- The most recent models officially released by well-known industry labs near the end of 2025, which we manually picked. This includes Allen AI's OLMo-3 (Olmo et al., 2025), NVIDIA's Nemotron nano (Blakeman et al., 2025) and cascade (Wang et al., 2025).

We release the evaluation results at https://huggingface.co/datasets/hlzhang109/proteus-2k. The most up-to-date part of proteus-2k, namely those built on new

20

| Base model family | # models |
|---|---|
| Mistral-7B-v0.3 (Jiang et al., 2023) | 143 |
| Qwen2.5-1.5B (Team et al., 2024b) | 124 |
| Llama-3.2-3B (Grattafiori et al., 2024) | 100 |
| gemma-2-9b (Team et al., 2024a) | 97 |
| Qwen3-4B-Base (Yang et al., 2025) | 90 |
| Qwen2.5-3B (Team et al., 2024b) | 89 |
| Llama-3.2-1B (Grattafiori et al., 2024) | 83 |
| gemma-2-2b (Team et al., 2024a) | 75 |
| Qwen2.5-32B (Team et al., 2024b) | 63 |
| Meta-Llama-3-8B (Grattafiori et al., 2024) | 62 |
| Llama-3.1-8B (Grattafiori et al., 2024) | 56 |
| Mistral-7B-v0.1 (Jiang et al., 2023) | 45 |
| Llama-2-13b-hf (Touvron et al., 2023) | 44 |
| gemma-3-1b-pt (Team et al., 2025) | 42 |
| SmolLM3-3B-Base (Allal et al., 2025) | 39 |
| Qwen2-7B (Yang et al., 2024) | 36 |
| Yi-34B (Young et al., 2024) | 34 |
| Others | 124 |
| **Total** | **1346** |

*Table 5.* Model counts by base model family among the newly evaluated models.

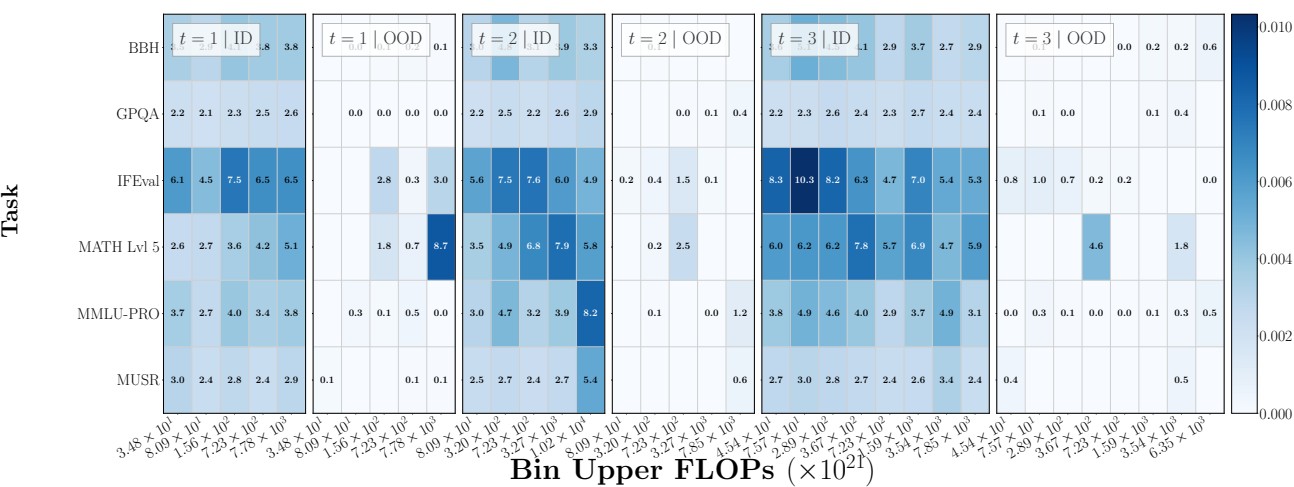

Figure 15. In-distribution and out-of-distribution pinball loss across log-compute bins for train ($\mathcal{P}_t$) and validation ($\mathcal{P}_{t+1}$), $t = 1, 2, 3$ across six Open LLM Leaderboard tasks.

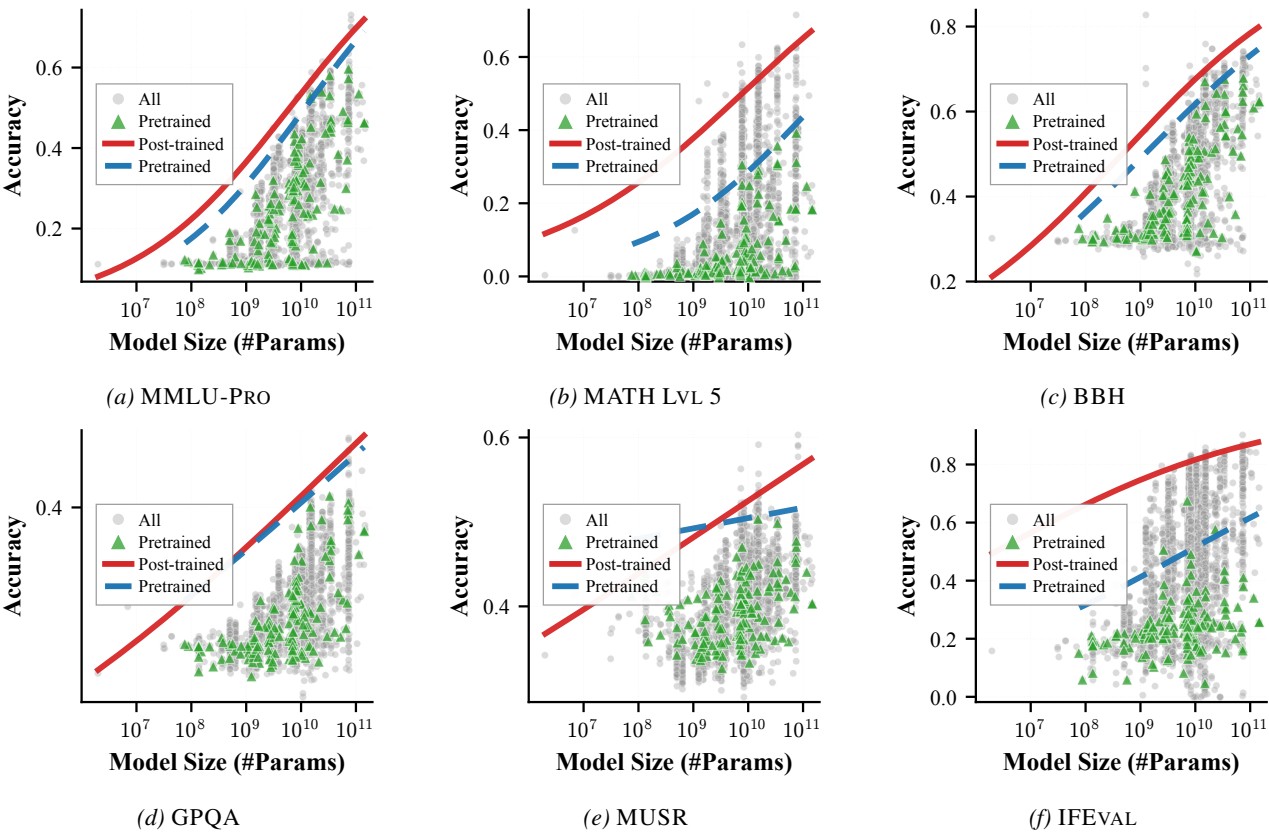

Figure 16. Sigmoid Capability Boundaries as functions of model size.

base models that do not appear on the Open LLM Leaderboard, is provided at https://huggingface.co/datasets/hlzhang109/proteus-selected.

The results for models released before the retirement of the Open

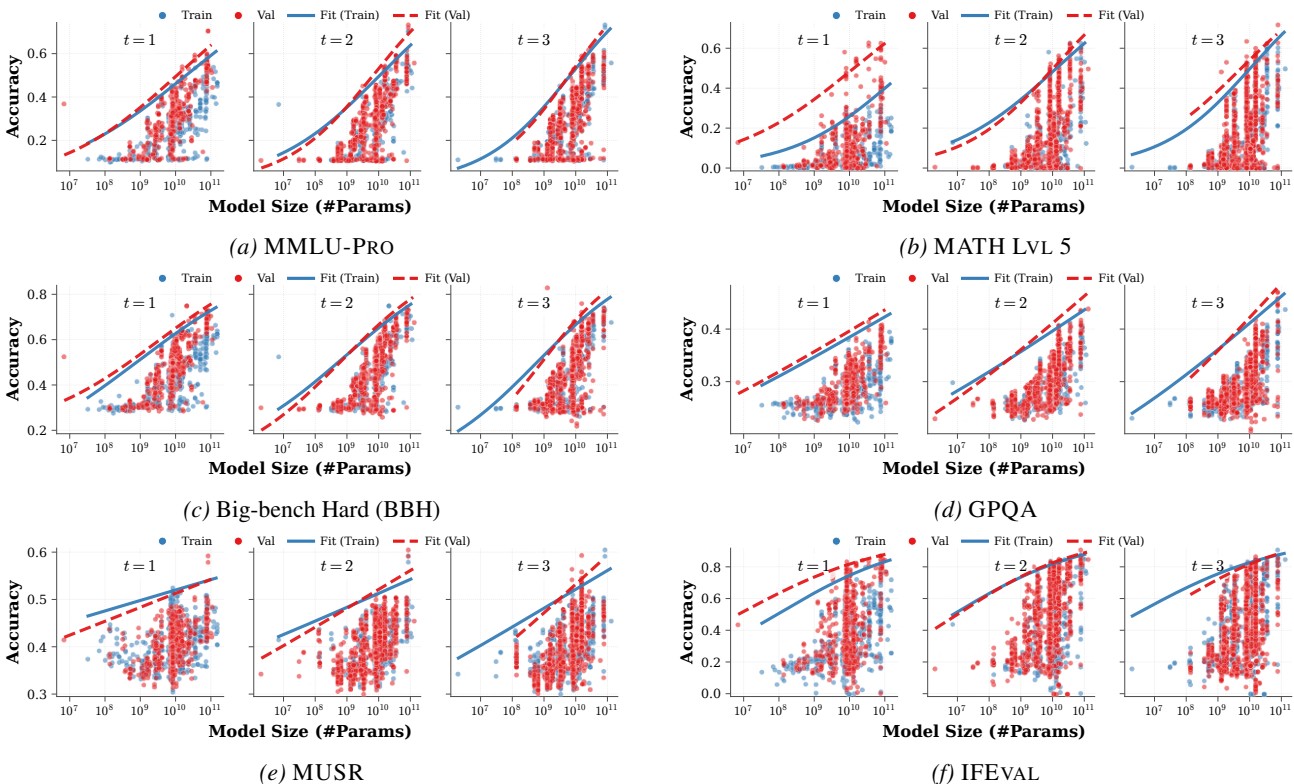

*Figure 17.* Comparison of sigmoid performance boundaries for $(\mathcal{P}_t)$ and $(\mathcal{P}_{t+1})$, $1 \leq t \leq 3$ across different tasks.

LLM Leaderboard are shown in Figure 18, while those of the later models are shown in Figure 19, where the models are additionally divided into two classes according to whether the base models are new.

Overall, we find that the capability boundaries that we estimate from the Open LLM Leaderboard reliably upper-bound the best possible accuracies attainable on various tasks, with only one caveat: on MATH LVL 5, there are several notable outliers in the right panel of Figure 19b.

## G. Sensitivity to Smoothed-Pinball Hyperparameters

Our sigmoid capability-boundary estimator in Section 2.1 minimizes a *smoothed* pinball objective with two numerical hyperparameters: the smoothing parameter $\kappa$ in $\ell_\tau$, and the ridge weight $\lambda$ in $\lambda\,\Omega(\theta)$. Throughout the main paper, we fix $\kappa = 50$ and $\lambda = 10^{-3}$. This appendix verifies that our empirical conclusions (e.g., the relative ranking of estimator families and the cross-temporal coverage patterns) are not artifacts of these choices.

**Why in-sample cross-validation is not needed in our setting.** We do *not* treat $(\kappa, \lambda)$ as model-selection hyperparameters in the usual sense. The sigmoid family is low-dimensional with explicit monotonicity/range constraints, so the dominant difficulty is *period shift* (fit on $\mathcal{P}_t$, evaluate on $\mathcal{P}_{t+1}$), rather than in-period overfitting. Moreover, $\kappa$ only controls how closely the smooth loss approximates the non-smooth check loss in a narrow band around zero residual, and $\lambda$ is included primarily for numerical conditioning rather than increased expressivity. Accordingly, cross-temporal evaluation already constitutes the intended validation; in-period cross-validation can add variance while optimizing for a different objective (within-period prediction).

As a sanity check, we ran an auxiliary tuning experiment: within each $\mathcal{P}_t$, we randomly split observations into two halves, tune $(\kappa, \lambda)$ over a small grid by minimizing validation pinball loss, and then re-evaluate on $\mathcal{P}_{t+1}$. Figure 20 shows that (i) selected hyperparameters concentrate in a narrow region (typically $\kappa \in \{20, 50\}$ and $\lambda \in \{10^{-4}, 10^{-3}\}$), and (ii) OOD

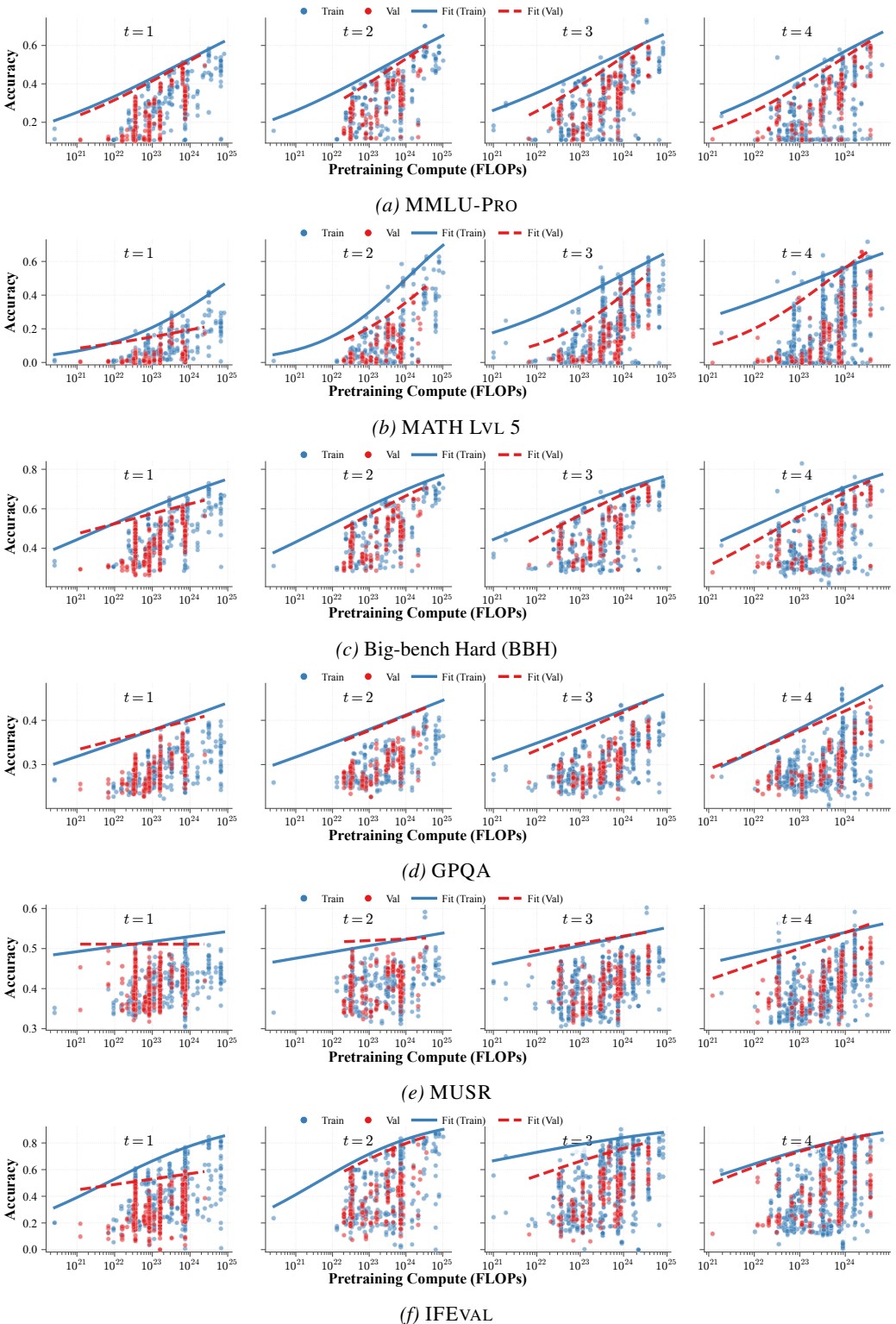

*Figure 18.* Comparison of sigmoid performance boundaries within each $\mathcal{P}_t$ for old and held-out new models.

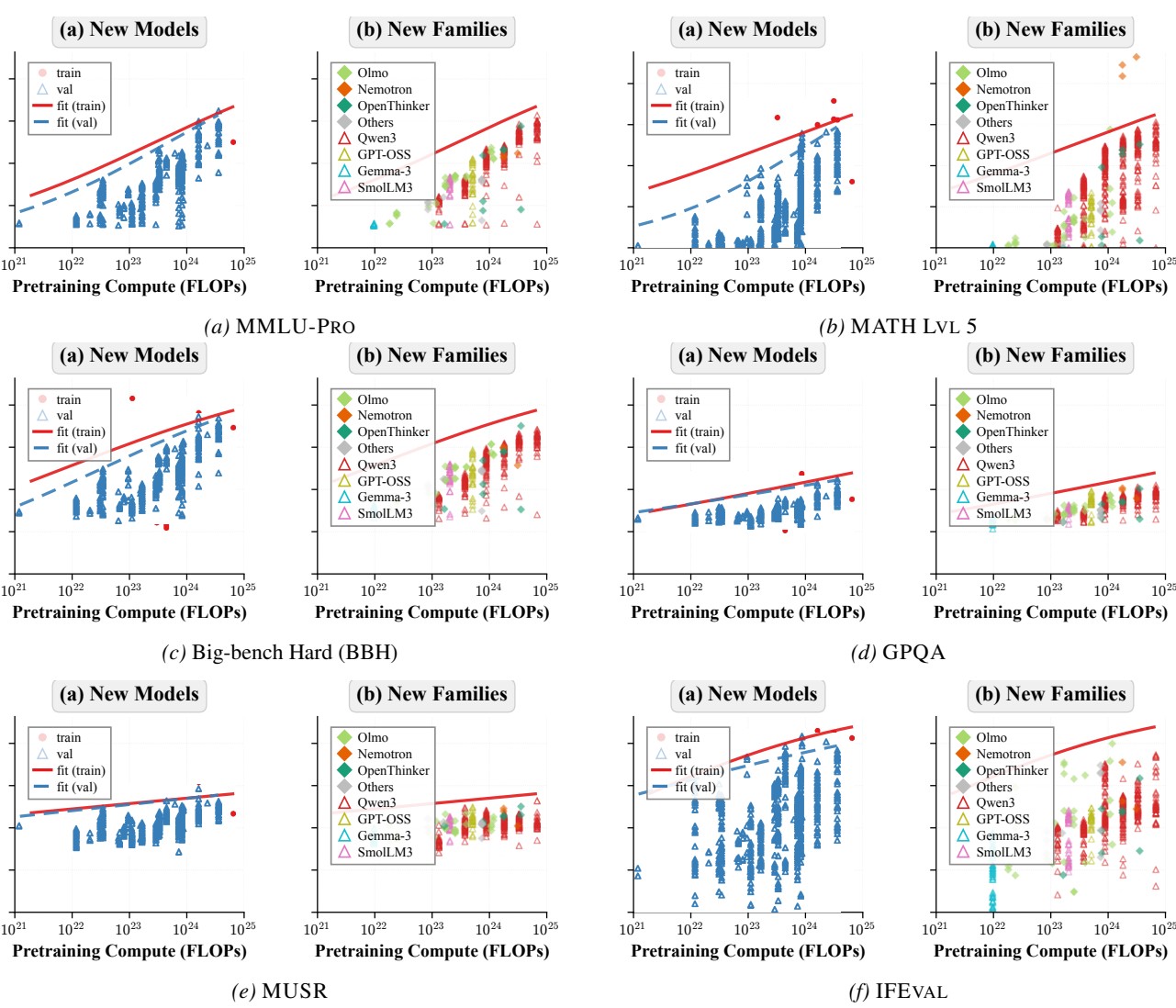

*Figure 19.* Comparison of sigmoid performance boundaries within each $\mathcal{P}_t$ for old and held-out new models

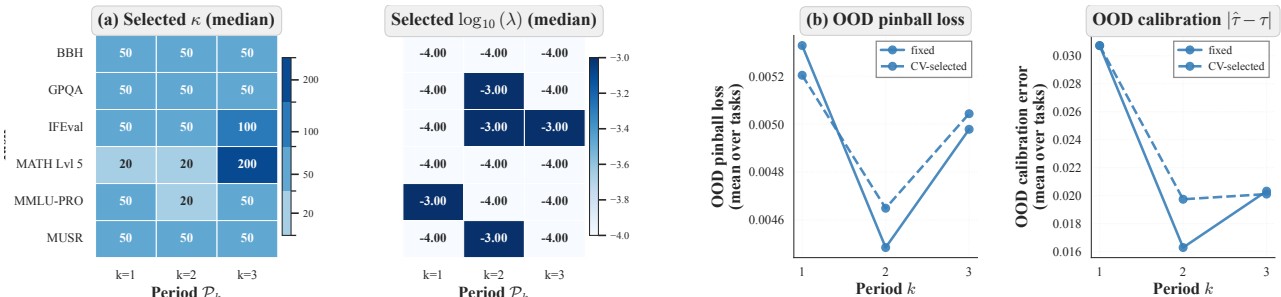

*Figure 20.* **In-period tuning has negligible effect on cross-temporal generalization.** The fixed default $(\kappa, \lambda) = (50, 10^{-3})$ used throughout the paper is competitive with values selected by random in-period splits. Left: $\kappa$ and $\log_{10} \lambda$ selected by in-period tuning (aggregated across random splits). Right: OOD metrics on $\mathcal{P}_{t+1}$: fixed default vs. in-period tuned $(\kappa, \lambda)$.

pinball loss and coverage are essentially unchanged relative to the fixed default $(\kappa, \lambda) = (50, 10^{-3})$. Thus, cross-validation provides little marginal benefit for the cross-temporal goal emphasized in the main paper.

**Grid sensitivity and practical recommendations.** We further swept $\kappa \in \{20, 50, 100, 200, 1000\}$ and $\lambda \in \{10^{-4}, 10^{-3}, 10^{-2}, 10^{-1}\}$ and measured OOD pinball loss and absolute coverage error on two representative tasks: BBH (typically well-calibrated in Section 3) and MATH LVL 5 (where temporal drift is most apparent). Figures 21 and 22 show that performance is stable across a broad "reasonable" region, but degrades for overly large ridge ($\lambda = 10^{-1}$). Within the stable region, $\kappa$ has a comparatively mild effect once it is moderate, while coverage can improve slightly for smaller $\lambda$.

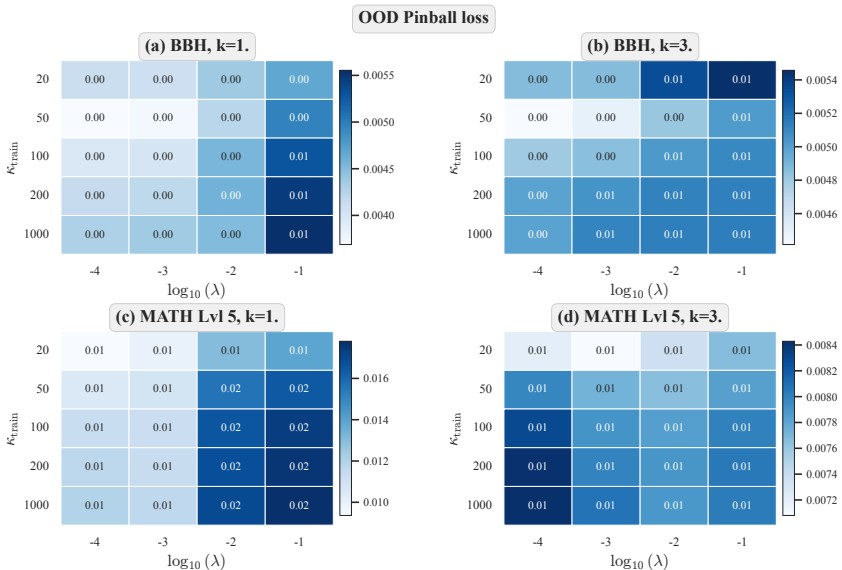

*Figure 21.* **OOD pinball loss under a $(\kappa, \lambda)$ sweep.** Lower is better. Results are shown for two tasks and two representative period splits ($t=1$ and $t=3$).

# H. Additional Results

## H.1. Public Leaderboards of Frontier Models

In this subsection, we apply the same methodology to fit a sigmoid scaling law using data from Epoch AI. Compared with the Open LLM Leaderboard, Epoch AI includes many closed-source models but the total number of evaluated models is smaller.

The results are shown in Figure 23. We can see that MATH Lvl 5 and Mock ATME show no pattern of performance gain from increasing the FLOPs. GPQA diamond, on the other hand, indicates a clear scaling in the FLOPs.

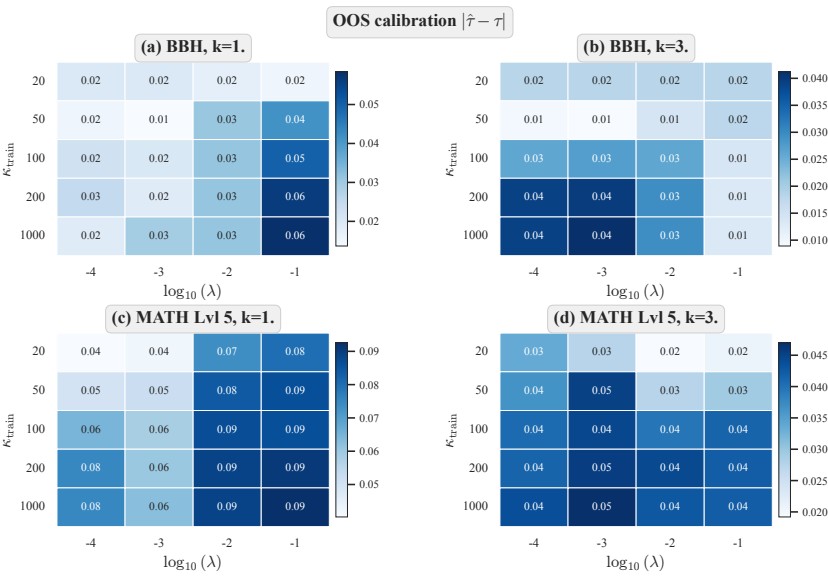

*Figure 22.* **OOD absolute coverage error** $|\hat{\tau} - \tau|$ **under a** $(\kappa, \lambda)$ **sweep.** Lower is better. Coverage is most sensitive to overly large $\lambda$, while $\kappa$ has a weaker effect once it is in a moderate range.

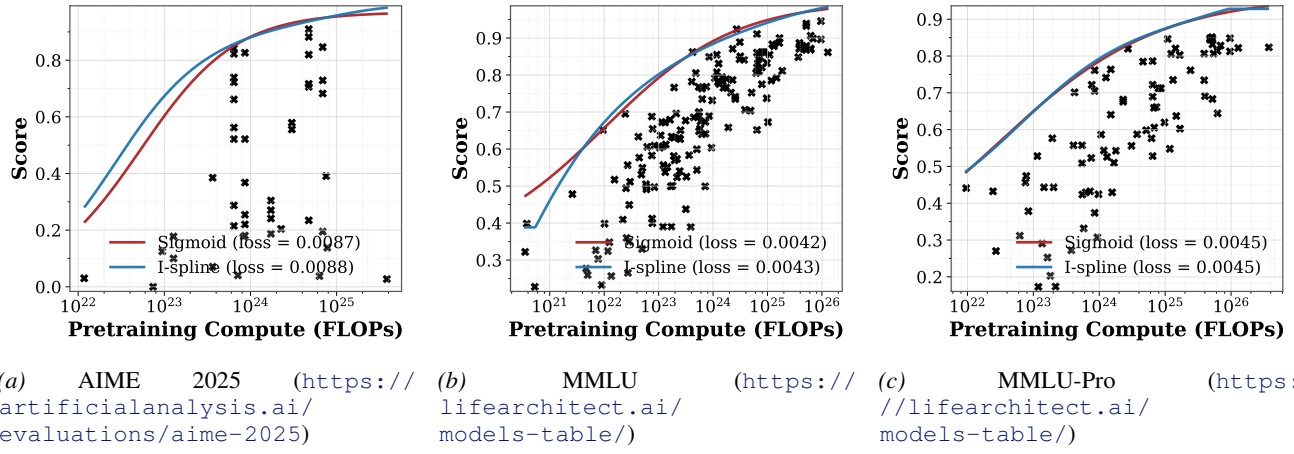

*(a)* AIME 2025 (https:// artificialanalysis.ai/ evaluations/aime-2025)

*(b)* MMLU (https:// lifearchitect.ai/ models-table/)

*(c)* MMLU-Pro (https: //lifearchitect.ai/ models-table/)

*Figure 23.* Sigmoid scaling law for frontier models. Evaluation data is publicly available from **Artificial Analysis** and **lifearchitect.ai**.

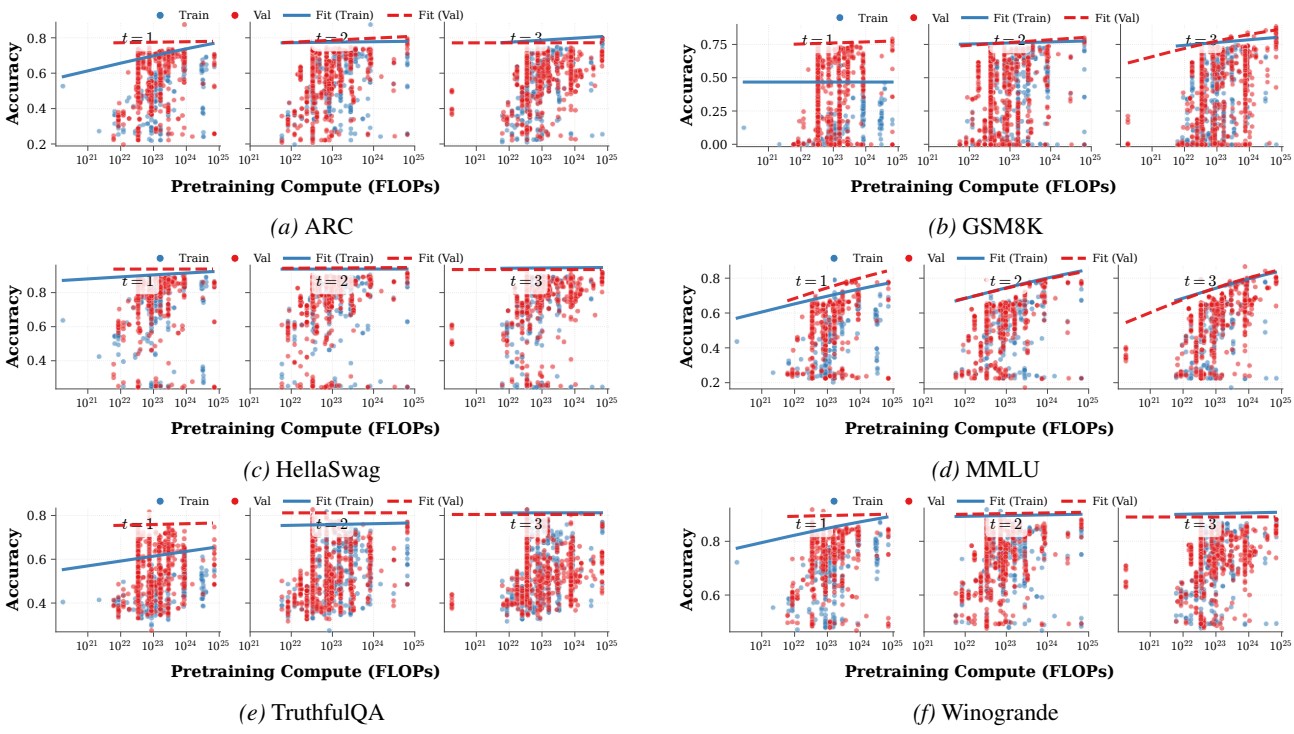

*Figure 24.* Comparison of sigmoid performance boundaries across different time periods $\mathcal{P}_t$ Open LLM Leaderboard v1.

## H.2. Results for Open LLM Leaderboard v1

In Figure 24 we present the sigmoid scaling laws learned on six tasks from the Open LLM Leaderboard v1, which is an older version compared with the v2 studied in the main part of the paper. We also define four time periods and investigate how model performance changes with both compute and time. The findings are quite different from that of v2: while several benchmarks such as GSM8K and TruthfulQA induces large performance gain when moving from $\mathcal{P}_1$ to $\mathcal{P}_2$, all benchmarks are saturated by $\mathcal{P}_3$. Furthermore, except from MMLU where a clear scaling relationship between FLOPS and capability boundary is observed, on remaining benchmarks larger FLOPs bring little or no performance gains in $\mathcal{P}_3$. This strongly indicates that these benchmarks were either too old or faced severe contamination issues, *even when the leaderboard was still active.*

## H.3. Latent Capability Factors and Prescriptive Boundaries

In this section, we provide more details about the PCA results mentioned in Theorem 3.2.

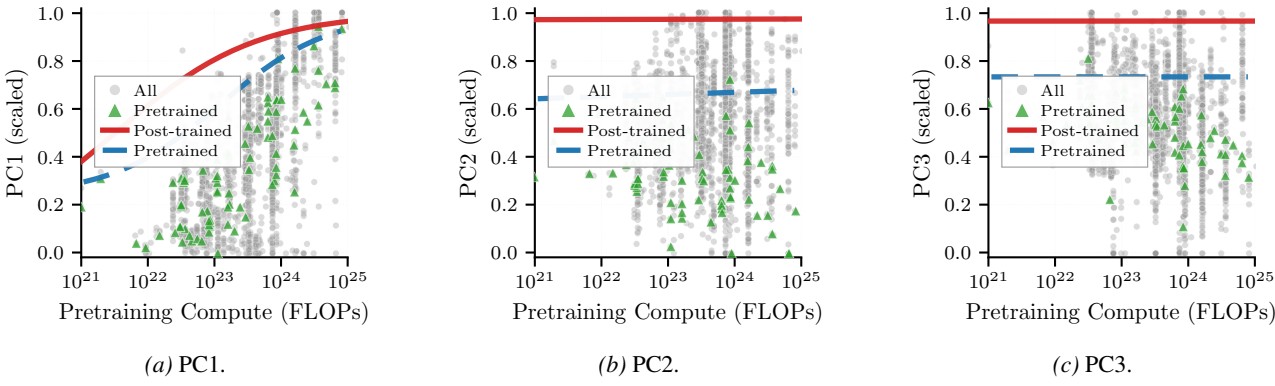

*Figure 25.* **The scaling of different principal components.**

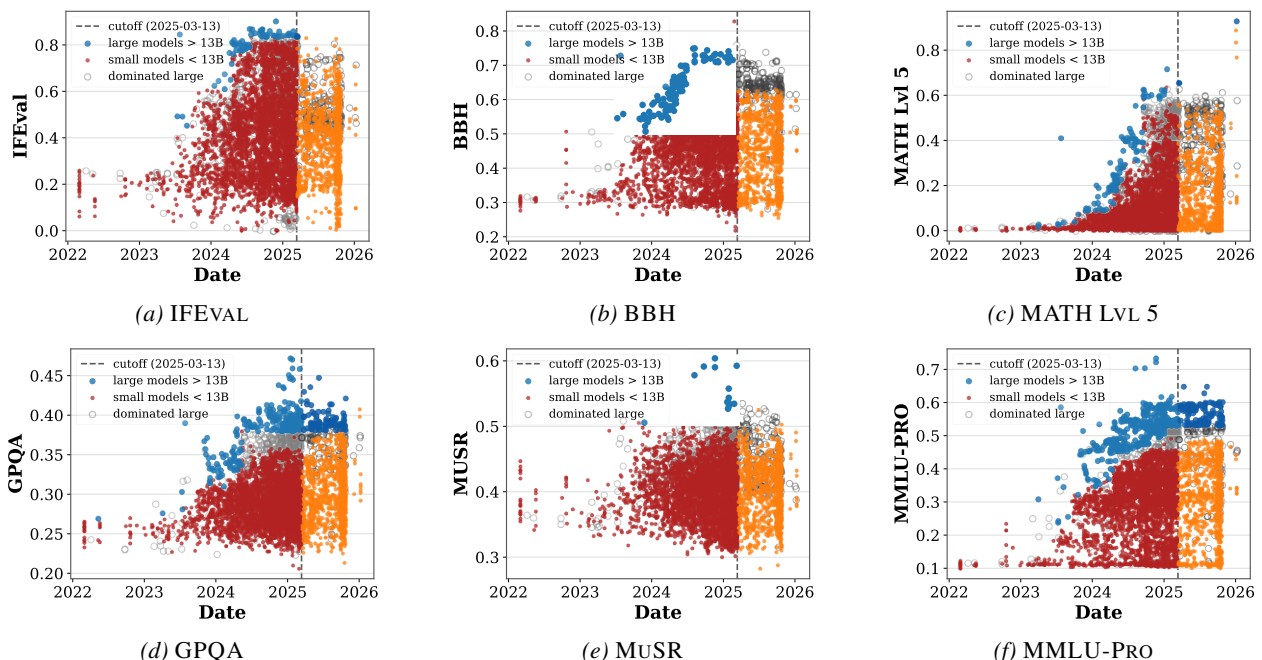

*Figure 26.* **Open LLM Leaderboard v2: saturation diagnostics by task.**

It is easy to see that while PC1 demonstrate a clear scaling in the compute, PC2 and PC3 have capability boundaries that are almost flat. This implies that the scaling law we established for the Open LLM Leaderboard may largely be attributed to advances in a single component.

## I. Saturation Analysis across Open LLM Leaderboard Versions and Tasks

This appendix provides the complete set of plots used to discuss saturation effects and the "slow death of scaling" narrative. We reproduce the core logic of Hooker (2025, Figure 3) on the Open LLM Leaderboard v1 and v2.

These plots are observational: they reflect submitted models, training recipes, post-training, and benchmark targeting over time. They should not be read as a controlled "parameter scaling law"; rather, they summarize how easily larger models translate into higher leaderboard scores for a given task.

**Open LLM Leaderboard v2.** Saturation is highly task-dependent on v2. In our runs, knowledge-heavy or knowledge+reasoning tasks (e.g., MMLU-Pro, GPQA) exhibit materially less domination by small models than pure reasoning tasks (e.g., MATH Lvl 5).

**Open LLM Leaderboard v1.** (Hooker, 2025) uses the (now-archived) v1 leaderboard suite; these plots show why conclusions about the "death of scaling" can be sensitive to the benchmark suite. Many v1 tasks show strong frontier convergence indicating a more saturated evaluation regime.

## J. Greedy Optimization for the Balanced I-Optimal Design

This appendix provides implementation details for the greedy solver used to approximately maximize the *balanced I-optimal* design criterion in each period (Section 4.1). The goal is to select a subset of candidate models $S_t \subseteq P_t$ under the per-period size budget $\sum_{i \in S_t} c_i \leq U_t$.

**Notation.** We reuse the balanced objective $\Phi_\lambda(S) = \Phi_{\text{info}}(S) + \lambda \, \Phi_{\text{bal}}(S)$ and the definitions of $\Phi_{\text{info}}$ and $\Phi_{\text{bal}}$ from Section 4.1. For the greedy updates, it is convenient to collect the following quantities:

- **Candidates and metadata.** For each $i \in P_t$, we assume we know $(z_i, c_i, b(i))$: log pre-training compute $z_i$, evaluation

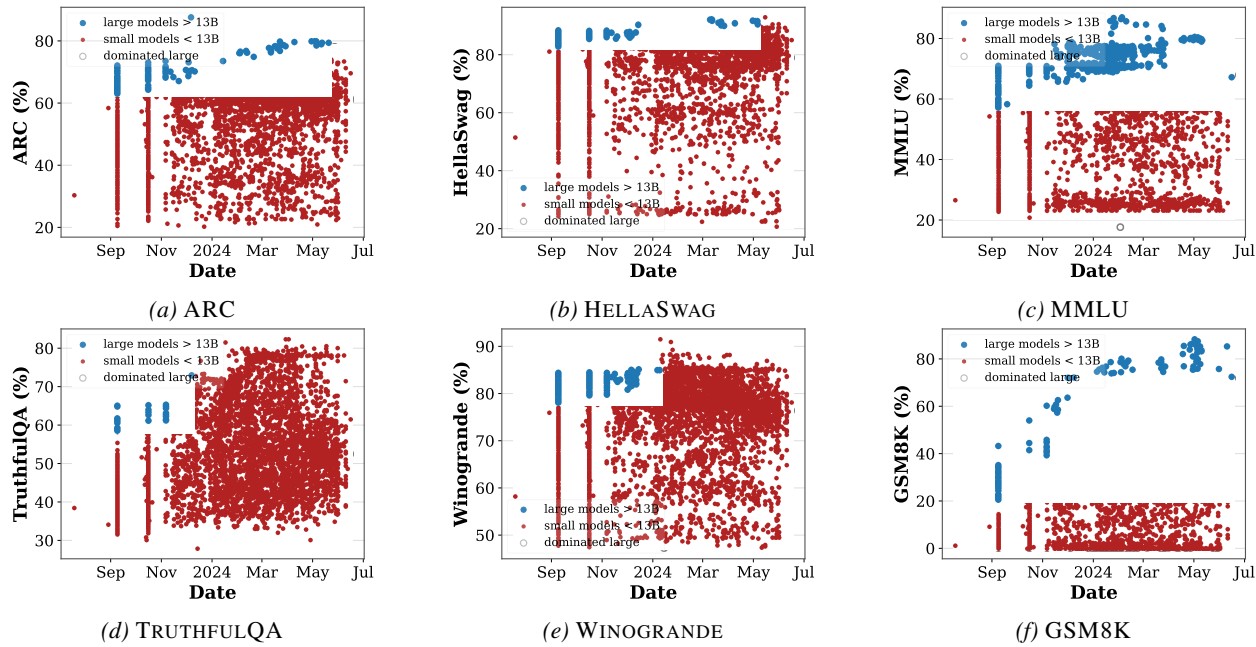

*Figure 27.* **Open LLM Leaderboard v1: saturation diagnostics by task.**

cost $c_i$, and log-compute bin index $b(i) \in \{1, \dots, B\}$ (Section 2.3).

- **Local Jacobians.** Let $\theta_0$ be a nominal frontier parameter obtained from an initial fit. Define $j_i \equiv j(z_i; \theta_0) \in \mathbb{R}^p$ as the Jacobian of the high-quantile boundary $q_\tau(z; \theta)$ with respect to $\theta$ evaluated at $\theta_0$ (explicit form in Section 4.1), where $p = 4$.

- **Bin midpoints.** Let $\{\tilde{z}_b\}_{b=1}^B$ be the bin midpoints and $j_b \equiv j(\tilde{z}_b; \theta_0)$ the associated Jacobians. We use weights $w_b$ (uniform $w_b = 1/B$ in our experiments) and define

$$A \equiv \sum_{b=1}^B w_b\, j_b j_b^\top \in \mathbb{R}^{p \times p}.$$

- **Inverse information.** For a current design $S$, let

$$K(S) \equiv \left( \eta I + \sum_{i \in S} j_i j_i^\top \right)^{-1},$$

where $\eta > 0$ is a small ridge term for numerical stability.

- **Bin counts.** Let $n_b(S) \equiv |\{i \in S : b(i) = b\}|$ and let $\varepsilon > 0$ be the balance constant in $\Phi_{\mathrm{bal}}$.

**Algorithm.** Algorithm 1 summarizes the greedy gain-per-cost procedure used in our experiments.

---

**Algorithm 1: Greedy optimization for the balanced I-optimal design**

**Require:** Period-$t$ candidates $P_t$ with metadata $\{(z_i, c_i, b(i))\}_{i \in P_t}$; budget $U_t$
**Require:** Bin midpoints $\{\tilde{z}_b\}_{b=1}^B$ and weights $\{w_b\}$ (e.g., $w_b = 1/B$)
**Require:** Nominal sigmoid parameters $\theta_0 = (y_0, L, a_0, b_0)$; ridge $\eta > 0$; balance constant $\varepsilon > 0$; tradeoff $\lambda \geq 0$
**Ensure:** Selected subset $S_t \subseteq P_t$ with $\sum_{i \in S_t} c_i \leq U_t$
1: **Precompute local geometry (at $\theta_0$).**
2: **for** each $i \in P_t$ **do**
3:     $j_i \leftarrow j(z_i; \theta_0)$                                         (Jacobian of $q_\tau(z; \theta)$ w.r.t. $\theta$)

4: **end for**
5: **for** $b = 1$ to $B$ **do**
6:     $j_b \leftarrow j(\tilde{z}_b; \theta_0)$
7: **end for**
8: $A \leftarrow \sum_{b=1}^{B} w_b\, j_b j_b^\top$ $\qquad\qquad\qquad$ (so $\Phi_{\text{info}}(S) = -\text{tr}(AK)$)
9: **Initialize with a small anchor set.**
10: Choose $S$ (e.g., two extreme-$z$ models and two near $z^\star = -a_0/b_0$); $U_{\text{rem}} \leftarrow U_t - \sum_{i \in S} c_i$
11: $K \leftarrow \left( \eta I + \sum_{i \in S} j_i j_i^\top \right)^{-1}$
12: $n_b \leftarrow |\{i \in S : b(i) = b\}|$ for $b = 1, \ldots, B$
13: **while** there exists feasible $i \in P_t \setminus S$ with $c_i \leq U_{\text{rem}}$ **do**
14:     **Evaluate gain-per-cost for each feasible candidate.**
15:     **for** each feasible $i \in P_t \setminus S$ with $c_i \leq U_{\text{rem}}$ **do**
16:         $u \leftarrow j_i; \quad v \leftarrow Ku; \quad \alpha \leftarrow 1 + u^\top v$
17:         $\Delta_{\text{info}}(i) \leftarrow \dfrac{v^\top A v}{\alpha}$ $\qquad\qquad\qquad$ (equivalently $-\text{tr}(AK_i) + \text{tr}(AK)$)
18:         $\Delta_{\text{bal}}(i) \leftarrow \log(n_{b(i)} + 1 + \varepsilon) - \log(n_{b(i)} + \varepsilon)$
19:         $g(i) \leftarrow \dfrac{\Delta_{\text{info}}(i) + \lambda\, \Delta_{\text{bal}}(i)}{c_i}$
20:     **end for**
21:     $i^\star \leftarrow$ feasible candidate with the largest $g(i)$
22:     **if** $g(i^\star) \leq 0$ **then**
23:         **break** $\qquad\qquad\qquad$ (no positive-gain addition remains)
24:     **end if**
25:     $u^\star \leftarrow j_{i^\star}; \quad v^\star \leftarrow Ku^\star; \quad \alpha^\star \leftarrow 1 + (u^\star)^\top v^\star$
26:     $K \leftarrow K - \dfrac{v^\star (v^\star)^\top}{\alpha^\star}$ $\qquad\qquad\qquad$ (Sherman–Morrison update)
27:     $S \leftarrow S \cup \{i^\star\}; \; U_{\text{rem}} \leftarrow U_{\text{rem}} - c_{i^\star}; \; n_{b(i^\star)} \leftarrow n_{b(i^\star)} + 1$
28: **end while**
29: **return** $S_t \leftarrow S$

---

Below, we briefly justify the key computations used in Algorithm 1 and document a few implementation choices.

**Sherman–Morrison update and closed-form gain.** Let $S$ be the current design and $i \notin S$ a candidate with Jacobian $u = j_i$. Write $K = (\eta I + \sum_{j \in S} j_j j_j^\top)^{-1}$ and $\alpha = 1 + u^\top Ku$. The rank-one update gives

$$K_i \equiv \left( \eta I + \sum_{j \in S \cup \{i\}} j_j j_j^\top \right)^{-1} = K - \frac{(Ku)(Ku)^\top}{\alpha}.$$

Using $\text{tr}(A(Ku)(Ku)^\top) = (Ku)^\top A(Ku)$, the marginal information gain admits the closed form

$$\Delta_{\text{info}}(i) = -\text{tr}(AK_i) + \text{tr}(AK) = \frac{(Ku)^\top A(Ku)}{1 + u^\top Ku},$$

which is what we compute in the inner loop (with $v \equiv Ku$). This avoids refactoring and reduces each candidate evaluation to $O(p^2)$ operations.

**Anchor initialization.** The greedy selection requires an initial set $S$ for which the local geometry is well-conditioned. In practice we initialize with a small anchor set that spans the observed compute range: two models near the minimum/maximum $z$, and (when available) up to two additional models near the nominal sigmoid inflection point $z^\star = -a_0/b_0$. We then fit an initial boundary to obtain $\theta_0$ and proceed with greedy additions.

**Optional 1-exchange "polish."** Greedy forward selection is fast but not guaranteed to reach a local optimum under the knapsack constraint. Optionally, after the greedy pass we apply a small number of Fedorov-style 1-exchange moves: remove one $j \in S$ and swap in a feasible $\ell \notin S$ if it increases $\Phi_\lambda$. Empirically this step yields only modest improvements, but it provides a robustness check on the greedy solution.

Figure 28 and 29 report how the resulting design quality varies with the budget parameter $\alpha$ (per-period and averaged over periods).

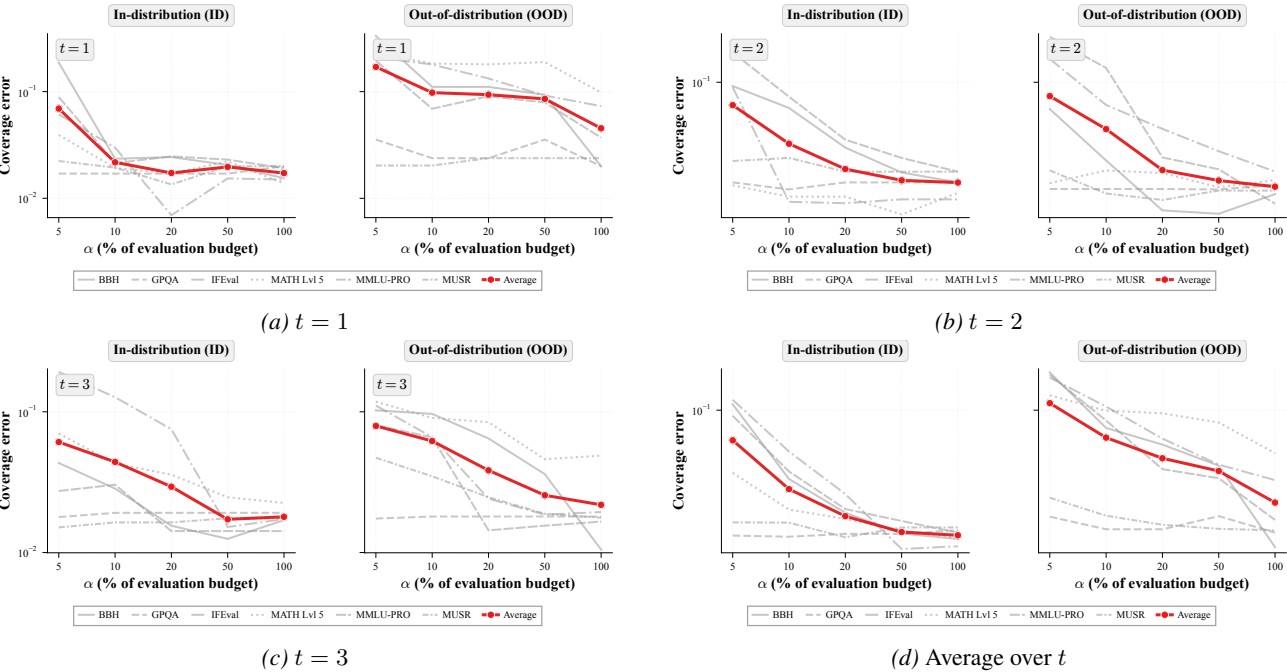

*Figure 28.* In-sample and out-of-sample *coverage calibration error* on period $t + 1$ as a function of budget parameter $\alpha$ when the boundary is estimated using balanced I-optimal design on period $t$. Curves correspond to different evaluation tasks.

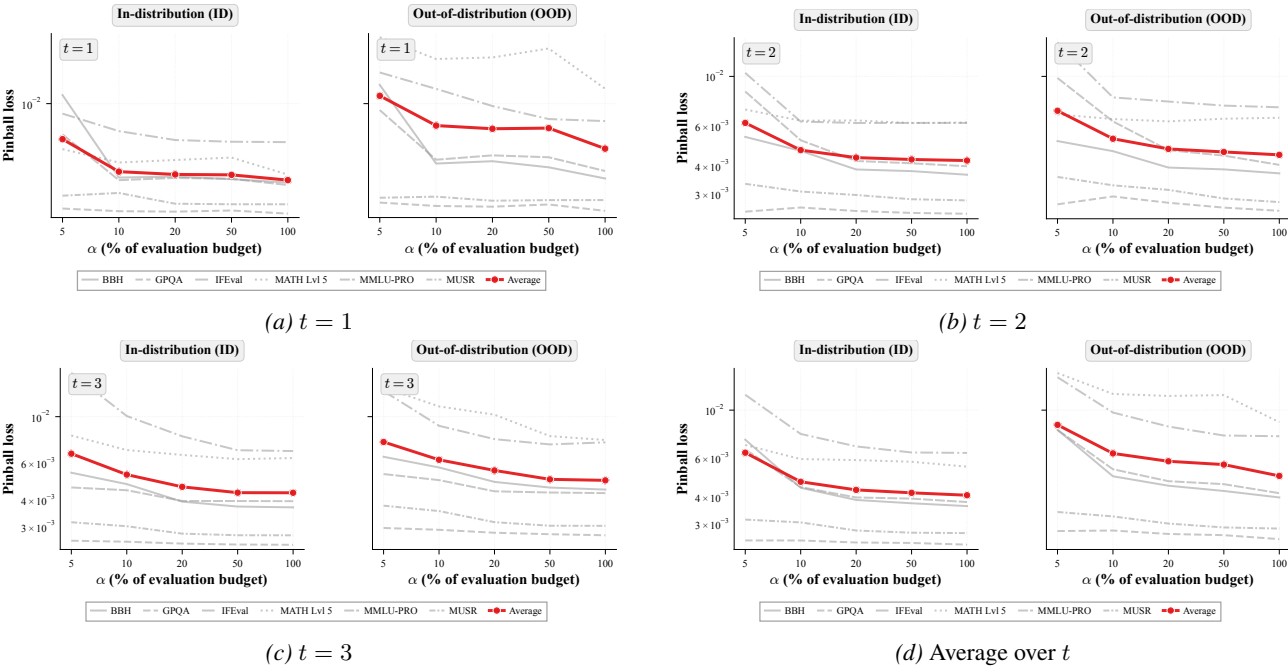

*Figure 29.* In-sample and out-of-sample pinball loss on period $t + 1$ as a function of budget parameter $\alpha$ when the boundary is estimated using balanced I-optimal design on period $t$. Curves correspond to different evaluation tasks.

