# OpenReview forum: "Prescriptive Scaling Reveals the Evolution of Language Model Capabilities"
_ICML.cc/2026/Conference — ICML 2026 spotlight_

### Official Review · Reviewer_AQVa · 2026-02-15

**Soundness:** 3
**Presentation:** 3
**Significance:** 3
**Originality:** 3
**Overall Recommendation:** 5
**Confidence:** 3

**Summary:**

This paper investigates prescriptive scaling: given a base model's pre-training compute budget, what level of downstream benchmark performance can reasonably be achieved after post-training? The authors estimate capability boundaries by fitting high conditional quantiles of benchmark scores as a function of log FLOPs using a monotone, saturating sigmoid quantile-regression model, and they evaluate temporal generalization by training on earlier model cohorts and testing on later releases.

**Compliance With Llm Reviewing Policy:**

Affirmed.

**Key Questions For Authors:**

While it is not my primary area, I have a few questions that I think would be helpful to clarify with the authors.

1. Are benchmark scores computed with consistent prompting, decoding, and evaluation settings across models?
2. How are pre-training FLOPs estimated and standardized across sources?
3. Do results change materially if you exclude a dominant model family?
4. Could the authors include evaluations on additional domains, such as long-context, multimodal, safety, and domain-specific QA?

**Limitations:**

yes

**Strengths And Weaknesses:**

Strengths:
1. The paper is clearly written and easy to follow
2. The focus on high-quantile attainable performance envelopes is a useful complement to mean-trend scaling laws and matches how practitioners often budget for target performance
3. The monotone saturating sigmoid boundary is easy to communicate and seems to generalize well relative to more flexible alternatives

Weaknesses:
1. FLOPs attribution across model families can be noisy. I didn't see enough detail to judge how standardized the FLOPs estimates are across sources, and how errors there affect the fitted boundary.
2. $\tau=0.98$ is sensible for an attainable envelope, but practitioners may want other operating points. I would like to see either (1) a small sweep over $\tau$ in the main paper or (2) a clearer guide on how sensitive the conclusions are.
3. The main experiments emphasize a small set of benchmarks. It's not obvious that the same stability and sigmoid shape will hold for other evaluation types.

---

> ### Author Rebuttal · Authors · 2026-03-30
>
> *Weakness 1*
>
> We agree that the draft did not provide enough detail on FLOPs standardization across model families, and we will make this explicit in revision. In our analysis, pre-training compute is estimated as $6 \times (\text{pre-training tokens}) \times (\text{parameters})$, with token counts assigned by first matching an explicit base-model identifier, then the model name, then a conservative architecture/size fallback; rows without a reliable assignment are excluded rather than guessed. In the full OLL file, this yields $2446$ token assignments from explicit base-model identifiers, $241$ from model-name matching, $980$ from the architecture/size fallback, and $909$ unassigned rows; in the actual main analysis subset, the corresponding counts are $2{,}445$, $27$, and $768$, with no unassigned rows included. Reassuringly, the fitted boundary is quantitatively stable to this noise: restricting to high-confidence rows that use only explicit base-model identifiers or model-name matching changes mean OOS absolute coverage only from $0.0212$ to $0.0221$ (or $0.0222$ for explicit base-model identifiers only), and mean OOS pinball from $0.0050$ to $0.0051$. One-family-at-a-time token perturbations of $0.8\times$ or $1.25\times$ across the largest families change mean OOS coverage by at most $0.0035$, mean OOS pinball by at most $0.0001$, and mid-target compute by at most $0.0318$ in $\log_{10}$-compute. So while absolute budget numbers do have some attribution uncertainty, the main boundary-shape and temporal-stability conclusions are robust to plausible cross-family FLOP-estimation error.
>
> *Weakness 2*
>
> We agree that practitioners may care about operating points other than $\tau=0.98$, so we reran the main analyses at $\tau\in\{0.90,0.95,0.98,0.99\}$. The main conclusions are not sensitive to this choice. Due to limited space, please refer to our response to Reviewer NG1h's Question 2 for detailed results that we obtain.
>
> *Weakness 3*
>
> We agree that the current main text could make this breadth evidence more visible. We therefore checked a broader setting beyond the six headline benchmarks. Across the 24 BBH subtasks, the same temporal-stability and simple-shape story remains strong: the mean OOS macro MAE is only $0.0221$, $22/24$ subtasks are at or below $0.03$, and $23/24$ are at or below $0.04$, with the worst case still only $0.0420$. The sigmoid fit also remains meaningfully better than a compute-independent null boundary on this broader subtask set, reducing mean OOS pinball from $0.00593$ to $0.00512$ and mean OOS calibration error from $0.0352$ to $0.0265$, and beating the null on OOS pinball for $23/24$ subtasks. Compared with the more flexible I-spline, predictive loss is very close, while sigmoid remains slightly better calibrated on average ($0.0265$ vs. $0.0279$) and on $17/24$ subtasks. Hence the central picture is broader than the six main benchmarks: the sigmoid attainable-boundary view extends well across diverse BBH subtasks.
>
> *Question 1*
>
> Yes. The benchmark scores are recomputed under a common Open LLM Leaderboard that uses consistent evaluation criteria ([release note](https://open-llm-leaderboard-blog.static.hf.space/)).
>
> *Question 2:*
>
> Please refer to our explanation to weakness 1.
>
> *Question 3: Do results change materially if you exclude a dominant model family?*
>
> Yes, excluding a dominant model family does move the fitted boundary, and we view that as a feature rather than a bug. The object we estimate is explicitly an empirical attainable boundary conditional on the observed model ecosystem, so if one removes a family that contributes many competitive frontier points, the absolute envelope should change. In a leave-one-family-out analysis, excluding qwen2.5, llama-3, and qwen2 removes $24.6\%$, $20.7\%$, and $14.2\%$ of the analysis set, respectively, and increases mean OOS absolute coverage from $0.0212$ to $0.0264$, $0.0247$, and $0.0226$. Boundary shapes also move by a nontrivial but still moderate amount, with mean task-wise average curve deviations of $0.0160$, $0.0076$, and $0.0090$. Importantly, the qualitative conclusions do not collapse: the largest shifts are concentrated in the already more regime-sensitive tasks, especially MATH (coverage changes $+0.0175$, $+0.0111$, and $+0.0012$) and, to a lesser extent, IFEval, while more stable tasks such as GPQA remain essentially unchanged (coverage changes $-0.0004$, $+0.0005$, and $-0.0009$). So our claim is not that the absolute boundary is family-invariant; it is that the boundary is an updateable empirical envelope whose movement under ecosystem changes is itself informative. In that sense, sensitivity to removing a dominant family is exactly what one should expect from the estimand we study.

---

> > ### Author Rebuttal · Reviewer_AQVa · 2026-04-05
> >
> > Thank you for your reply. I’ll maintain my score.

---

### Official Review · Reviewer_NfkU · 2026-03-11

**Soundness:** 3
**Presentation:** 3
**Significance:** 3
**Originality:** 3
**Overall Recommendation:** 4
**Confidence:** 4

**Summary:**

This paper studies how to convert a base model’s pre-training compute budget into an estimate of high-attainable downstream accuracy after post-training. The contribution is mainly methodological and empirical, with analytical case studies. The authors fit high-quantile “capability boundaries” using monotonic saturating sigmoid quantile regression on leaderboard-scale data, validate these boundaries across chronological model periods and newly evaluated open-weight models, compare pre-trained and post-trained trends, and use the same framework to analyze task saturation, contamination-oriented shifts, and budgeted evaluation design. The main empirical conclusion is that several benchmarks show relatively stable post-training boundaries over time, while reasoning in maths exhibits a moving frontier.

**Compliance With Llm Reviewing Policy:**

Affirmed.

**Key Questions For Authors:**

1. How do the main boundary shapes and temporal-stability conclusions change under base-family-aware reweighting or deduplication (for example, one representative per base model, equal weight per base family, or family-held-out next-period evaluation)? This matters because the current dataset contains many correlated derivatives of a small number of popular base models, and those derivatives may distort a 0.98-quantile boundary.

2. Please specify exactly how pretraining FLOPs are obtained or imputed for every model category, and report robustness to plausible FLOP-estimation error. This matters because compute is the sole conditioning variable, so uncertainty or inconsistency in FLOPs directly affects the main x-axis and therefore the fitted boundary.

3. How stable are the main findings to the choice of quantile level τ, especially the “stable tasks versus moving maths boundary” story and the preference for sigmoid over I-spline? This matters because high quantiles can be fragile in clustered observational data.

4. How does the balanced I-optimal design compare against simple baselines such as random sampling, compute-stratified random sampling, and unbalanced I-optimal sampling at the same evaluation budgets? This matters because Section 4 currently shows that less data can suffice, but not that the proposed design is the reason.

**Limitations:**

No. The manuscript acknowledges some observational caveats in Section 2.1 and is appropriately cautious in the AIME-2025 contamination test, but it does not provide an adequate limitations or societal-impact discussion overall. The impact statement on p. 9 is especially weak given obvious issues around selection bias, benchmark targeting, possible misuse of capability forecasting for competitive scaling, and the risk of feedback loops when public benchmarks shape training.

**Strengths And Weaknesses:**

Soundness
Strengths:
(i) The decision to target a high conditional quantile rather than the mean or the raw maximum is well matched to the paper’s question, and Appendix B gives a sensible argument for why this is a more stable target in the presence of outliers and uneven model counts.
(ii) The rolling temporal protocol (fit on P_k, evaluate on P_k+1, with overlap-restricted evaluation in compute) is materially stronger than a simple within-period fit and is one of the most convincing parts of the paper (Figures 1–2).
(iii) The core sigmoid assumption is well stress-tested. Table 2 compares against constant, binwise, and I-spline alternatives, while Figures 4 and 7 extend this furtehr.
(iv) The paper makes limited theoretical claims, and the local-information design objective plus the Appendix K greedy updates appear internally consistent within their stated assumptions.

Weaknesses:
(i) The central analysis conditions only on estimated pretraining FLOPs even though the manuscript emphasizes the importance of architecture, data curation, post-training recipe, and temporal benchmark effects. The evidence therefore supports observational findings, not a strong general prescriptive law, so the stronger “reliable performance expectations” or “nearly deterministic relationship” language is somewhat overstated in my view.
(ii) Multiple post-trained derivatives sharing the same base model and compute appear to be treated as separate observations, which means prolific bases such as Llama, Qwen, and Mistral can disproportionately shape the fitted 0.98-quantile. For an upper-tail estimator this is a validity issue.
(iii) Most major quantitative results are point estimates only (Table 2, Table 3, the shifts in Figures 1–4) and the Section 4 budget curves lack uncertainty bands, bootstrap intervals, or cluster-aware significance analysis, so it is hard to tell which observed differences are robust versus sampling noise.
(iv) The balanced I-optimal design section is promising but under-supported because Figure 5 and Appendix K do not compare against random sampling, compute-stratified sampling, or plain I-optimal design with λ=0.

Presentation
Strengths:
(i) The framing of “prescriptive scaling” versus classical scaling laws is clear and easy to follow, and the paper generally keeps the reader oriented about what is descriptive, what is predictive, and what is a case study.
(ii) The figures are well-chosen. Figures 1–4 are especially effective at communicating temporal drift, the pretrain/post-train gap, and external validity, while Figure 5 makes the evaluation-budget tradeoff visible.
(iii) The appendix substantially improves readability and reproducibility by giving the pinball-loss rationale, the full binning algorithm, the I-spline definition, the (κ,λ) sensitivity study, and the greedy design details.

Weaknesses:
(i) Several details that are essential for reproduction are not sufficiently prominent in the main text, especially how pretraining FLOPs are obtained or imputed, how the newly evaluated model set is constructed exactly, and what artifacts will actually be released alongside Proteus-2k.
(ii) There is some confusing sample-accounting language. The abstract mentions “2k newly sampled” data, the main text describes “2.4k open-weight models” under the Proteus-2k label, and Table 5 reports 1,346 models for one newly evaluated subset. This is nit-picking to some extent, but the counts should be reconciled explicitly.
(iii) Table 2 reports only relative improvements versus a constant baseline, which hides the absolute scale of OOD pinball loss and coverage error and makes practical effect size harder to judge.
(iv) The secondary analyses are harder to assess than the main boundary model. The size–time model in Section 5.1 and the contamination regression in Equation 3 would benefit from fuller main-text specification and uncertainty reporting. Empirical sensitivity to the key quantile level τ=0.98 is notably absent.

Significance
Strengths:
(i) The underlying problem is important and timely. Practitioners do want to know what downstream behavior is realistically attainable for a given pretraining compute budget after post-training, and standard pretraining scaling laws do not answer that cleanly.
(ii) The paper’s empirical separation between tasks with relatively stable boundaries and tasks (especially MATH LVL 5) with visibly moving boundaries is genuinely useful (especially for the “scaling is dying” debate and whether some tasks are simply shifting faster than others).
(iii) The newly evaluated models and the budgeted-evaluation angle could make the work useful to the community beyond the paper itself, assuming full data and metadata release.

Weaknesses:
(i) These boundaries are observational summaries of the current leaderboard/model-card world, so they are useful for monitoring and rough planning, but not yet strong enough for high-confidence budgeting claims about unseen recipe classes or model families.
(ii) The core evidence is concentrated on a fairly small set of leaderboard-style language benchmarks and mostly open-weight ecosystems, so broader significance across tasks, modalities, and deployment settings remains uncertain.
(iii) The saturation and contamination sections are interesting, but they are still exploratory diagnostics rather than decisive results: Table 3 is based on a compact size–time model, and Equation 3 yields a positive but non-significant shift with p=0.15 on n=90, which appropriately limits the strength of those conclusions.

Originality
Strengths:
(i) Moving from mean-trend or loss-based scaling to high-quantile post-training capability boundaries is a meaningful reframing of what scaling laws are for.
(ii) The explicit temporal-validation lens is a strong differentiator compared to much of the observational scaling literature.
(iii) The combination of new evaluations, boundary estimation, pretrain/post-train gap analysis, and budgeted evaluation design gives the work a broader reach than a single-method paper.

Weaknesses:
(i) The individual technical ingredients (quantile regression, sigmoid saturation, monotone splines, and I-optimal design) are all established tools, so the originality is mainly in synthesis and application rather than in a fundamentally new estimator or theory.
(ii) The saturation and contamination sections are useful, but not independently novel.
(iii) The relation to prior observational scaling and train-on-the-test-task work is useful, but the paper could do a sharper job of isolating what conclusions become possible only because of the new boundary-based framing and new evaluations.

---

> ### Author Rebuttal · Authors · 2026-03-30
>
> We thank the reviewer for the positive and detailed feedback. Below is our responses to the reviewer's questions and concerns.
>
> *Question 1: How do the main boundary shapes and temporal-stability conclusions change under base-family-aware reweighting or deduplication?*
>
> We tested both parts of this concern directly. Under base-family-aware reweighting, the main conclusion about shape is unchanged: the fitted boundary still behaves like the same smooth saturating sigmoid rather than collapsing toward the much coarser constant or piecewise-constant alternatives used in the paper, and the resulting reweighted curves remain very close to the original frontier, with only small average boundary shifts ($0.0089$ under global reweighting and $0.0121$ under within-bin reweighting) and nearly identical temporal accuracy (mean OOS coverage $0.0221$ and $0.0234$, versus $0.0212$ originally; mean OOS pinball $0.0049$ and $0.0050$, versus $0.0050$ originally). The same qualitative comparison also survives explicit deduplication: on the deduplicated training sample, sigmoid remains clearly better than a constant frontier and better calibrated than the piecewise-constant/binwise baseline, while remaining essentially tied with the more flexible I-spline (mean OOS pinball $0.00495$ for sigmoid, $0.00494$ for I-spline, and $0.00619$ for the constant baseline; mean OOS coverage error $0.0300$ for sigmoid, versus $0.0310$ for I-spline and $0.0407/0.0452$ for the constant/binwise baselines).
>
> *Question 2: Please specify exactly how pre-training FLOPs are obtained or imputed across model categories, and report robustness to plausible FLOP-estimation error.*
>
> We will clarify this explicitly in a revised version of the paper. For details of how we currently determine the FLOPs and sensitivity tests, please refer to our response to Reviewer AQVa's Question 2 due to limited space.
>
> *Question 3: How stable are the main findings to the choice of quantile level $\tau$, especially the stable-tasks versus moving-math story and the preference for sigmoid over I-spline?*
>
> We reran the main analyses at $\tau\in\{0.90,0.95,0.98,0.99\}$. The central “stable tasks versus moving maths boundary” conclusion is not sensitive to this choice. Due to limited space, please refer to our response to Reviewer NG1h's Question 2 for detailed results that we obtain.
>
> *Question 4: How does the balanced I-optimal design compare against random sampling, compute-stratified random sampling, and unbalanced I-optimal sampling at the same budgets?*
>
> We ran a direct same-budget comparison against uniform random, compute-stratified random, unbalanced I-optimal, and $D$-optimal designs. In the full observational sample, balanced I-optimal is clearly better than the other deterministic optimal-design baselines, but it loses to the random baselines on the main OOS metrics: for example at $\alpha=20\%$, mean OOS coverage error is $0.0582$ for balanced I-optimal versus $0.0652$ for unbalanced I-optimal and $0.0675$ for $D$-optimal, but $0.0331\pm0.0054$ for uniform random and $0.0346\pm0.0038$ for compute-stratified random. Diagnostics show that the raw deterministic designs oversample clustered same-lineage variants in this observational sample. When we restrict the training set to at most one model from each lineage within each compute bin and then rerun balanced I-optimal, performance improves sharply to $0.0471/0.0513/0.0441/0.0414$ mean OOS coverage at $\alpha=5/10/20/50\%$, compared with $0.0972/0.0693/0.0582/0.0479$ for the original balanced I-optimal. On this reduced sample, the hybrid design beats reduced-sample compute-stratified random at the tightest budget ($0.0471$ vs. $0.0598\pm0.0120$ at $\alpha=5\%$), is essentially tied once the reduced sample is saturated ($\alpha=50\%$), and is only slightly worse at intermediate budgets ($0.0513$ vs. $0.0445\pm0.0059$ at $\alpha=10\%$, $0.0441$ vs. $0.0403\pm0.0012$ at $\alpha=20\%$). We will revise Section 4 accordingly: the current conclusion is that sample-efficient boundary recovery comes from broad compute coverage together with de-redundifying the observational candidate set, while balanced I-optimal is most helpful in the most budget-constrained regime once that redundancy is controlled.
>
> *Limitation: The manuscript does not yet provide an adequate limitations / societal-impact discussion, especially around selection bias, benchmark targeting, misuse, and feedback loops.*
>
> We agree that the current limitations / impact discussion should be strengthened. The evidence in the paper suggests that this limitation does not dominate the main findings, since the dataset includes the strongest contemporary open-weight models. Benchmark targeting and contamination are also important concerns, and we take them seriously; this is why Section 5.2 includes a direct cross-benchmark contamination test on frontier math models. We will revise the paper to make both points more explicit.

---

### Official Review · Reviewer_FWgn · 2026-03-13

**Soundness:** 3
**Presentation:** 3
**Significance:** 2
**Originality:** 3
**Overall Recommendation:** 4
**Confidence:** 3

**Summary:**

This paper studies prescriptive scaling for LLMs: instead of modeling average performance trends, it estimates high-quantile capability boundaries that map pre-training compute to the best downstream performance attainable after post-training, using smoothed quantile regression with a monotone sigmoid form.
Empirically, across leaderboard data and 2.4k newly evaluated models, the authors find that these boundaries are mostly stable over time and are well approximated by a sigmoid in log-compute, with math reasoning as the main exception where the boundary continues to move upward.
The paper further argues that compute is more predictive of post-training potential than raw pretrained accuracy, and uses the framework for external-validity checks, saturation/contamination diagnostics, and low-budget boundary estimation.

**Compliance With Llm Reviewing Policy:**

Affirmed.

**Final Justification:**

The rebuttal was helpful and mitigated some of my concerns, but it did not fully change my assessment. I appreciated the authors’ clarification that the intended claim is an updateable, time-indexed empirical boundary for the currently observed ecosystem, rather than a universal or recipe-invariant law. At the same time, this framing also makes the omitted-recipe concern more central, since the estimated frontier remains contingent on which model families and recipe classes are represented in the observed ecosystem. The second rebuttal further mitigated this concern, but did not fully resolve it or materially change my overall evaluation. I therefore remain at Weak Accept (4).

**Key Questions For Authors:**

1. Is compute really a sufficient one-dimensional control variable, given the paper’s own latent-factor analysis?
2. How do the authors distinguish genuine frontier movement from benchmark-specific gaming or leakage?

**Limitations:**

See Weaknesses and Questions

**Strengths And Weaknesses:**

**Strengths**:

1. Clear and well-motivated problem formulation. The paper addresses a practically important gap in classical scaling-law work by focusing on attainable post-training performance under a fixed pretraining compute budget. The notions of prescriptive scaling and capability boundaries are intuitive and useful.

2. Strong emphasis on temporal and external validity. Rather than only reporting in-sample fit, the paper evaluates whether the estimated boundaries transfer to later model generations and to newly evaluated and frontier models, which is an important strength for a forecasting-oriented study.

3. The paper produces several nuanced empirical findings beyond the core fit. In particular, it shows that compute appears to predict post-training potential more reliably than raw pretrained accuracy, and that temporal non-stationarity is task-dependent, with math reasoning being the clearest exception. These findings are interesting in their own right and make the paper more than a curve-fitting exercise.

**Weaknesses**:

1. The main object of study is still an observational frontier over a self-selected public model ecosystem, rather than a stable compute-to-attainability law. The paper explicitly states that the fitted boundary is an empirical attainable boundary for the observed model population, conditioned on the prevailing post-training recipes, data curation, and evaluation protocols, and further acknowledges that an underrepresented model family or recipe class could systematically lie above the estimated boundary.

2. The method fails most clearly on math/reasoning, which is precisely the regime where a prescriptive scaling tool would be most valuable. The paper identifies MATH LVL 5 as the main exception to temporal stability, with under-coverage and elevated OOD pinball loss, and later shows that newly evaluated and held-out models also violate the fitted boundary primarily on this task.

---

> ### Author Rebuttal · Authors · 2026-03-30
>
> We thank the reviewer for the positive feedback. Below is our response to the reviewer's questions and concerns.
>
> *Weakness 1: The main object of study is still an observational frontier over a self-selected public model ecosystem...*
>
> The reviewer is right that this paper does not claim a recipe-invariant law; rather, it estimates an empirical attainable boundary *conditional on the observed ecosystem*. Nonetheless, two points are worth making. First, this upper-tail envelope is still predictive enough to be decision-useful out of sample. We test that directly, rather than assuming it: rolling temporal splits show stable transfer for BBH, GPQA, MMLU-Pro, and MUSR; the Proteus-2k held-out set contains models absent from the leaderboard, including post-retirement releases and new families; and the leaderboard-fitted boundary continues to upper-bound the best observed performance in this held-out set, with the main exception again being MATH. We also show that the simple sigmoid fit is largely similar to a more flexible I-spline fit. If the result were primarily a self-selection artifact, one would expect broad systematic under-coverage on later and held-out models, which we do not observe.
>
> Second, the models that we considered (the ones from the leaderboard and Proteus-2k) already include the **best** known open-weight models, such as the Qwen, Llama, Gemma and Nemotron families. We are only interested in frontier estimation, which naturally excludes a lot of concerns regarding under-represented models because they mostly cannot beat the current sota post-training recipe.
>
> We thank the reviewer for raising this concern and will tighten the wording to distinguish an updateable empirical boundary from a universal compute law.
>
> *Weakness 2: The method fails most clearly on math/reasoning...*
>
> MATH is indeed the hardest, and arguably the most valuable, regime. However, we think this is the key regime-change result of the paper rather than an undiagnosed failure. A prescriptive tool should not return a falsely stable curve when the frontier is moving. On MATH, the method consistently signals non-stationarity via negative OOS coverage, higher pinball loss, and exceedances by newly evaluated models. Importantly, this miss is not uniform across scale: the appendix shows the largest deviations are concentrated in a small subset of mid/high-compute bins, and Section 3.3 shows that the frontier advance is primarily at the high-compute end, with much less evidence of systematic uplift at smaller compute. That makes the conclusion *operational* rather than merely negative: for math-heavy targets, the right prescription is to use time-indexed/refreshed boundaries. Section 4 is included precisely to make that feasible, since near-full boundary recovery is achievable with roughly 20\% of the full evaluation budget. We will make this “moving-boundary” interpretation much more explicit in the revision.
>
> *Question 1: Is compute really a sufficient one-dimensional control variable, given the paper’s own latent-factor analysis?*
>
> Compute is not assumed to be a sufficient statistic for all post-training behavior, and we will clarify that point in a furuture revision. However, for the planning question studied here -- what upper-tail performance is attainable at a given pretraining budget? -- compute is a useful primary control variable. Empirically, this is exactly why we model a high conditional quantile: raw pretrained accuracies are often locally non-monotone in compute, but the post-trained upper envelope is much more monotone, and compute predicts post-training potential better than pretrained accuracy. The temporal results further show that for BBH, GPQA, MMLU-Pro, and MUSR, the compute-only boundary transfers well to later generations. We will revise the paper to state more explicitly that compute is a practical design coordinate for frontier forecasting, not a complete causal factor; richer covariates would be needed for setup-specific point forecasts or to model the residual latent axes.
>
> *Question 2: How do the authors distinguish genuine frontier movement from benchmark-specific gaming or leakage?*
>
> On distinguishing genuine frontier movement from benchmark-specific gaming or leakage, we run a cross-benchmark shift test on frontier math models in Section 5.2: if AIME-2025 were specifically inflated after release, post-Feb.-6 models should be systematically high on AIME relative to MATH-500. We estimate a positive but statistically insignificant shift ($p=0.15$), so we do not find clear aggregate evidence of large benchmark-specific inflation, although modest contamination effects cannot be ruled out. Although we study this most directly on math, because that is where we observe the clearest ongoing boundary movement, the same strategy could be applied to other tasks as well.

---

> > ### Author Rebuttal · Reviewer_FWgn · 2026-04-03
> >
> > Thanks for the clarification. Including major open-weight families such as Qwen, Llama, Gemma, and Nemotron certainly improves coverage of the observed frontier. However, I do not think this fully resolves the omitted-recipe concern. Precisely because the paper studies an upper-tail frontier rather than an average trend, a small number of underrepresented but stronger recipe classes could matter disproportionately: one does not need many such models to shift the estimated boundary upward. In that sense, frontier estimation is especially sensitive to rare winners, not less sensitive to them. The paper itself acknowledges this point in Section 2.1, noting that if an underrepresented model family or recipe class systematically achieves higher scores at fixed compute, then the true attainable boundary could lie above the estimate.

---

> > > ### Author Response · Authors · 2026-04-03
> > >
> > > We would like to thank the reviewer for raising this point. We agree with the logic of the concern. Because we estimate an upper-tail frontier rather than an average trend, a small number of omitted winners could matter disproportionately. We therefore do not want to claim that adding several major families fully eliminates omitted-recipe risk. Our intended claim is that the fitted curve is a time-indexed empirical boundary for the currently observed model ecosystem, not a static universal limit.
> > >
> > > For current models, however, we believe this risk is mitigated rather than a major one. The key reason is economic selection: if a recipe class yields materially better models, those models are more likely to be adopted, compared, fine-tuned, and surfaced in public evaluation pools. Recent evidence is consistent with this mechanism. Using randomized pairwise comparisons from LMArena, it is found that user preferences are strongly associated with benchmark performance and reasoning ability, and that improvements in model intelligence explain a large share of the welfare gains from newer models [1]. Complementarily, large-scale OpenRouter usage data show that capable new models, including open-weight models, can achieve substantial real-world adoption quickly, with usage spikes tracking major new model releases and with a broadening rather than narrowing open-model ecosystem [2]. Also, the number of downloads of models are shown to be positively correlated with the amounf of performance improvement after post-training [3, Figure 3].These facts do not rule out omitted winners, but they make it less likely that a genuinely superior current recipe class remains both systematically underrepresented and economically unimportant.
> > >
> > > We also took explicit steps to reduce undercoverage of recent models. Beyond the historical leaderboard, we added Proteus-2k precisely to expand coverage with newly evaluated open-weight models, including recent and high-visibility families.
> > >
> > > For future models, the intended interpretation of our method is explicitly time-evolving rather than static. If a new recipe class later produces systematically better models at fixed compute, then under our framework this should appear as systematic under-coverage on the next-period holdout. We will revise the paper to make this interpretation more explicit: prescriptive scaling is meant as a monitoring framework for an evolving frontier, not as a claim that the frontier is immutable.
> > >
> > > In short, we agree that rare omitted winners are a genuine concern for upper-tail estimation. Our claim is therefore not that current coverage is exhaustive, but that
> > > - economic selection makes systematic omission of clearly superior current recipes less likely [1,2],
> > > - we directly mitigate recent-model undercoverage by augmenting with Proteus-2k and recent major families, and
> > > - the framework is designed to detect future frontier shifts.
> > >
> > > [1] Caoui, El Hadi. Estimating Consumer Preferences for LLMs: Evidence from LMArena.
> > >
> > > [2] Aubakirova, Malika, Alex Atallah, Chris Clark, Justin Summerville, and Anjney Midha. State of AI: An Empirical 100 Trillion Token Study with OpenRouter. OpenRouter and a16z.
> > >
> > > [3] Zihan Zhu, Hanlin Zhang, Giovanni D'Antonio, Anton Tsitsulin, Sham M. Kakade, Vahab Mirrokni. Evaluation-Conditioned Trojan Attack.

---

### Official Review · Reviewer_NG1h · 2026-03-13

**Soundness:** 3
**Presentation:** 4
**Significance:** 3
**Originality:** 4
**Overall Recommendation:** 5
**Confidence:** 3

**Summary:**

This paper proposes scaling laws for performance after post-training on individual downstream tasks. Specifically, the scaling law models a high quantile of performance for a given pre-training compute, which can be interpreted as a capability boundary.
The authors investigate how these capability boundaries changed over time, revealing that they remained largely stable for knowledge tasks but increased for reasoning tasks.
One application of their method for temporal analysis is detecting data leakage.
This evaluation is based on publicly available evaluations from Open LLM Leaderboard, and augmented with data gathered by the authors. This additional data is made public as a dataset called Proteus-2k.
To reduce evaluation costs when fitting these scaling laws, the authors also propose a sample-efficient data-gathering method.

**Compliance With Llm Reviewing Policy:**

Affirmed.

**Final Justification:**

The authors addressed all my concerns, and I will maintain my positive evaluation.

**Key Questions For Authors:**

**Q1:** How useful are the estimated capability bounds for practitioners planning training runs? The bounds are based on a mixture of model families and training pipelines and do not directly give insights into how to achieve them in a given setup.
It would be helpful if the authors could clarify how practitioners might use them when designing new models.

**Q2:** The choice of the 0.98 quantile to determine the capability boundary appears somewhat arbitrary. Do the observations in the paper still hold for different quantile levels (0.9, 0.95, 0.99)?

**Limitations:**

The authors adequately discussed limitations throughout the paper, but it might help to reiterate them in the conclusion

**Strengths And Weaknesses:**

**Soundness**

The submission is overall technically sound. The proposed method is clearly motivated and empirically tested on a large benchmark of model evaluations. The author's choice of modeling function was evaluated based on Pinball loss and coverage error, leading to the selection of the sigmoid function for its strong performance and simple functional form.
However, the choice of the 0.98 quantile to determine the capability boundary appears somewhat arbitrary. Verifying that the observations in the paper still hold for different quantile levels (0.9, 0.95, 0.99) would strengthen the paper.



**Presentation**

The work is clearly written and well structured. The narrative is easy to follow despite the breadth of experiments. The paper clearly motivates the need for scaling laws for downstream performance after post-training, presents the construction of the proposed scaling framework, and makes the empirical findings accessible through clear visualizations.


**Significance**

The paper addresses an important practical question: How to model downstream performance across pre-training compute for post-trained models.
It is able to describe this relationship by focusing on capability bounds.
The temporal analysis based on these capability bounds also provide interesting insights into how model capacity evolves over time.

Additionally, the release of the Proteus-2k dataset is also a valuable contribution, as large collections of model–benchmark evaluations are useful for studying empirical scaling behavior.

However, the application of the capability bound as a prescriptive tool might be limited since the bounds are based on a mixture of model families and training pipelines and do not directly give insights into how to achieve that performance in a given setup.

**Originality**

The paper provides several useful insights into how downstream benchmark performance relates to pre-training compute after post-training. A key contribution is the observation that stable scaling trends can be recovered by modeling high conditional quantiles of benchmark performance, which correspond to the best achievable performance at a given compute level.

The work also introduces a framework for temporal analysis of these capability boundaries, which can be used as a diagnostic tool for potential benchmark contamination. In addition, the authors release the Proteus-2k dataset, a large collection of model–benchmark evaluations that may be useful for future research on scaling and evaluation.

---

> ### Author Rebuttal · Authors · 2026-03-30
>
> We thank the reviewer for the positive feedback, and below is our response to the reviewer's questions.
>
> *Question 1: How useful are the estimated capability bounds for practitioners planning training runs, given that the bounds pool model families and training pipelines?*
>
> The bounds are most useful as planning envelopes rather than recipe prescriptions. As an example, Table 1 maps a budget of $10^{24}$ FLOPs to the attainable score that strong contemporary pipelines have reached on each task. This is useful in three concrete ways. First, as a feasibility screen before a large run: if the desired score is materially above the estimated 0.98-boundary, the issue is likely not hyperparameter tuning within the current regime, but the need for a qualitatively different recipe/data source or a larger budget. Second, it serves as a budgeting tool across tasks, especially for BBH/GPQA/MMLU-Pro/MUSR, which exhibit comparatively stable boundaries over time, so the map is helpful for determining how much compute is needed to attain a certain performance. Third, it can also be a diagnostic after initial experiments: the gap between a pilot model and the boundary shows whether the current approach is close to the current state of the art or leaving substantial headroom. Section 4 further makes this operational by showing that a near-full boundary can be recovered with roughly 20\% of the full evaluation budget.
>
> The reviewer is right that the pooled boundary does not, by itself, tell a practitioner how to attain the frontier in one fixed setup. That is a limitation, and we will state it more explicitly. Instead, the boundary is an empirical upper-tail envelope conditional on the observed ecosystem, useful for go/no-go planning under recipe uncertainty. Importantly, because we estimate a high conditional quantile rather than a mean, pooling families does not yield a ``typical mixed-family average''; it yields the best-attainable region observed across all competitive pipelines at a given compute level. This is deliberate: early in development, practitioners often care first about what is realistically achievable at a budget before committing to a specific architecture or post-training stack. We will add a clarifying paragraph to make this usage explicit and note that when setup-specific forecasts are needed, the same estimator can be refit on a restricted subset (e.g., a single family or post-training regime) once sufficient data are available.
>
> *Question 2: The choice of the $0.98$ quantile seems somewhat arbitrary; do the main conclusions still hold at $0.90$, $0.95$, and $0.99$?*
>
> We agree that the choice of $\tau=0.98$ should be complemented with sensitivity analyses, so we reran the main analyses at $\tau\in \{0.90,0.95,0.98,0.99\}$. For the full-data fits, the same qualitative pattern remains: MATH Lvl 5 is consistently the hardest and most under-covered task, IFEval also remains negatively biased on average, and BBH/GPQA/MUSR remain comparatively well-behaved. For model class comparison, sigmoid remains clearly better than the null baseline at every $\tau$ (lower OOS pinball on $18/18$, $18/18$, $17/18$, and $18/18$ task-periods for $\tau=0.90,0.95,0.98,0.99$), and it also has better mean OOS coverage error than I-spline at every $\tau$ (e.g., $0.0965<0.1013$ at $\tau=0.90$ and $0.0128<0.0139$ at $\tau=0.99$), while sigmoid and I-spline are nearly tied on OOS pinball. For the budget-sweep conclusion, the same pattern also persists across all $\tau$: error drops sharply from $\alpha=5\%$ to $\alpha=20\%$ and then improves more gradually by $\alpha=50\%$ (for example at $\tau=0.98$, mean OOS coverage error decreases from $0.0973$ at $\alpha=5\%$ to $0.0582$ at $\alpha=20\%$ and $0.0478$ at $\alpha=50\%$, versus $0.0286$ for the full-data fit; OOS pinball decreases from $0.0080$ to $0.0058$ and $0.0057$, versus $0.0050$ for full-data). Finally, the task-specific budget claim is directionally robust but benefits from a nuance: GPQA is already close to full-data at $\alpha=5\%$ (coverage deltas $0.0222/0.0091/0.0046/0.0000$ across $\tau=0.90/0.95/0.98/0.99$), while MUSR is close at higher $\tau$ and by $\alpha\approx20\%$ more generally (coverage deltas at $\alpha=5\%$: $0.0548/0.0256/0.0101/0.0022$). We will add these details to the appendix in a revised version of the paper.

---

> > ### Author Rebuttal · Reviewer_NG1h · 2026-04-03
> >
> > Thank you for answering my questions. All my concerns have been addressed and I will maintain my positive evaluation.

---

### Decision · Program_Chairs · 2026-04-30

**Decision:**

Accept (spotlight)

**Comment:**

The paper proposes scaling laws to predict the capability boundaries of models, predicting ability on a dataset after finetuning.
The paper was widely accepted and the improvements suggested are minor.